# Hypermethylation of gene body CpG islands predicts high dosage of functional oncogenes in liver cancer

Maria Arechederra[1], Fabrice Daian [1], Annie Yim[2], Sehrish K. Bazai[1], Sylvie Richelme[1], Rosanna Dono[1], Andrew J. Saurin [1], Bianca H. Habermann [1] & Flavio Maina [1]

Epigenetic modifications such as aberrant DNA methylation reshape the gene expression repertoire in cancer. Here, we used a clinically relevant hepatocellular carcinoma (HCC) mouse model (*Alb-R26$^{Met}$*) to explore the impact of DNA methylation on transcriptional switches associated with tumorigenesis. We identified a striking enrichment in genes simultaneously hypermethylated in CpG islands (CGIs) and overexpressed. These hypermethylated CGIs are located either in the 5′-UTR or in the gene body region. Remarkably, such CGI hypermethylation accompanied by gene upregulation also occurs in 56% of HCC patients, which belong to the "HCC proliferative-progenitor" subclass. Most of the genes upregulated and with hypermethylated CGIs in the *Alb-R26$^{Met}$* HCC model undergo the same change in a large proportion of HCC patients. Among reprogrammed genes, several are well-known oncogenes. For others not previously linked to cancer, we demonstrate here their action together as an "oncogene module". Thus, hypermethylation of gene body CGIs is predictive of elevated oncogene levels in cancer, offering a novel stratification strategy and perspectives to normalise cancer gene dosages.

[1] Aix Marseille Univ, CNRS, Developmental Biology Institute of Marseille (IBDM), Parc Scientifique de Luminy, Aix Marseille Univ, 13009 Marseille, France. [2] Computational Biology Group, Max Planck Institute of Biochemistry, 82152 Martinsried, Germany. Correspondence and requests for materials should be addressed to F.M. (email: flavio.maina@univ-amu.fr)

Appropriate timing and dosage of gene expression in healthy cells is ensured by complex processes integrating genetic and epigenetic information. Alterations of these mechanisms are frequent in cancer and underline functional changes in genes acting as oncogenes or tumour suppressors[1–3]. The use of high-throughput sequencing has contributed considerably to our understanding on how epigenetic modifications switch genomic regions from an inaccessible closed conformation to an open state–and vice-versa–contributing to changes in the transcriptome landscape[4–6]. DNA methylation is an essential epigenetic mechanism influencing gene expression levels in cells and alterations lead to dramatic changes in malignant cells. The cancer landscape is generally characterised by a diffuse DNA hypomethylation and by focal hypermethylation in CpG-rich regions known as CpG islands (CGIs)[1,7]. CGI hypermethylation at promoters represses transcription of genes acting as tumour suppressors, a well-known mechanism operating in cancer[8]. However, DNA methylation at intergenic regions and gene bodies is gaining relevance for its impact on gene expression[9,10]. Aberrant DNA methylation of large clusters of transcriptional enhancers, known as super-enhancers, leads to dramatic transcriptional changes of gene sets in cancer[11]. A large fraction of DNA methylation is also observed in gene body CGIs, with an apparent intriguing positive correlation between methylation and gene expression[12,13]. Such contradiction on DNA methylation effects in promoter versus gene body CGIs remains poorly understood.

The relevance of epigenetics in tumorigenesis has been further emphasised through recent large-scale screen analyses focused on cancer patients carrying either histone mutations or alterations in genes regulating DNA methylation–histone modifications[2]. Results from these studies highlighted how such mutations dramatically modify the epigenetic and gene expression landscapes. For example, aberrant DNA methylation has been recently reported in acute myeloid leukaemia patients with DNMT3A mutations[14]. Abnormal recruitment of PRC2 complex and DNA methylation occurs in paediatric glioblastoma with Histone H3 mutant variants[15]. Gene expression changes caused by histone H3K36 mutation is associated with sarcomagenesis[16]. Nevertheless, the epigenetic reshape occurs also in the absence of specific mutations in chromatin modulators[17]. It is the case of classical oncogenes and tumour suppressors, which can trigger profound chromatin alterations with consequences on gene expression[18,19]. For example, an oncogenic splice variant of EGFR leads to genome-wide activation of putative enhancers in glioblastoma[20]. Oncogenic EGFR leads to DNA methylation-mediated transcriptional silencing of tumour suppressors in lung cancer and glioblastoma[21]. Deregulated Ras signalling reshapes the enhancer landscape leading to aberrant oncogene expression[22]. PI3K/Akt pathway activation induces promoter-associated gene activation in breast cancer[23]. Overall, such screen approaches have also contributed to identify new genes, whose functional relevance in cancer was previously unknown and/or which deregulations can be used as cancer biomarkers for prognosis/patient stratification.

We recently reported a cancer mouse model in which slight increases in wild-type Met receptor tyrosine kinase (RTK) levels in the liver are sufficient for spontaneous tumours in mice (Alb-R26[Met]). These genetic studies conceptually illustrate how the shift from physiological to pathological conditions results from perturbations in subtle signalling dosage. Through gene expression analysis, the Alb-R26[Met] mice were shown to model a HCC patient subgroup corresponding to the so-called "proliferative-progenitor" subclass[24], demonstrating the clinical relevance of this genetic system. The uniqueness of this genetic system was also illustrated by its usefulness to identify new synthetic lethal interactions as potential therapies for HCC subgroups[24]. Here, we employed the Alb-R26[Met] cancer model for integrative genome-wide studies combining methylome and transcriptome outcomes and compared them with those from HCC patients. Results show an enrichment in genes overexpressed and with hypermethylated CGI, with expression levels positively correlating with the CGI distance to the ATG. Whereas most of the upregulated genes are well-known oncogenes, the implication of others in cell tumorigenic properties is demonstrated here through functional studies. Enrichment of genes both overexpressed and with hypermethylated CGIs characterises the "proliferative-progenitor" HCC patient subset, which is modelled by the Alb-R26[Met] genetic system. Collectively, these results show that an epigenetic reprogramming process ensuring increased dosage of an "oncogenic module" involving multiple genes operates in tumorigenesis.

## Results

**Alb-R26[Met] tumours recapitulate DNA methylation changes of HCC patient subgroups.** We recently showed how the Alb-R26[Met] genetic system is a unique tool to model: (a) the tumorigenic program, (b) the "proliferative-progenitor" HCC patient subgroup and (c) functionality of signalling alteration for drug discovery[24]. For its use to study the contribution of epigenetic modifications linked to cancer, we reasoned that it was first necessary to determine whether the Alb-R26[Met] tumorigenesis occurs in a stable genomic context or is associated with chromosomal deletions/duplications. Comparative genomic hybridisation analyses on DNA inputs from 16 Alb-R26[Met] tumours and 8 control livers excluded chromosomal instability (Supplementary Fig. 1). These findings therefore reinforce the appropriateness of the Alb-R26[Met] cancer model as a relevant genetic system to study the epigenetic reprogramming associated with cancer, which we addressed by bioinformatically integrating data from methylome and transcriptome screens (Fig. 1a).

DNA methylation changes were scored by performing Methyl-MiniSeq EpiQuest sequencing on 10 Alb-R26[Met] tumours (previously histologically identified as HCC[24]) and 3 control livers (Supplementary Fig. 2A). Mean methylation levels were modestly, yet significantly, different across all measured CpGs (P-value = 2.4E−03; Fig. 1b), being able to group tumours and controls into two distinct clusters (Fig. 1c). A remarkable predominance of global hypomethylation was observed in tumours compared with livers (Fig. 1b, Supplementary Fig. 2B). Accordingly, we observed an enrichment in hypomethylated CpGs located outside CGIs (P-value = 3E−04; Fig. 1b, Supplementary Fig. 2C). In contrast, a significant enrichment of hypermethylation at CpGs located within CGIs characterised Alb-R26[Met] tumours compared with control livers (P-value = 3.9E−03; Fig. 1b, d, Supplementary Fig. 2D). These traits of CpG methylation changes, according to the CpG location with respect to CGIs, are consistent with those largely reported in the literature[1]. Focusing on differentially methylated CpGs located at annotated CGIs, we identified 513 CGIs with a β-value methylation difference of ±0.2 and a false discovery rate (FDR) <0.05 (Fig. 1d, Supplementary Fig. 2D, Supplementary Data 1). These CGIs were homogeneously distributed amongst all 19 autosomal and 1 sex chromosome mouse pairs (Supplementary Fig. 2E). Among CGIs with differentially methylated CpGs, 82% were hypermethylated in Alb-R26[Met] HCC compared to controls (Fig. 1d).

To explore the relevance of these methylation changes in the context of human HCC disease, we used genome-wide DNA methylation data from a cohort of 41 HCC patients, for which data are available for both: (a) methylation and expression; (b)

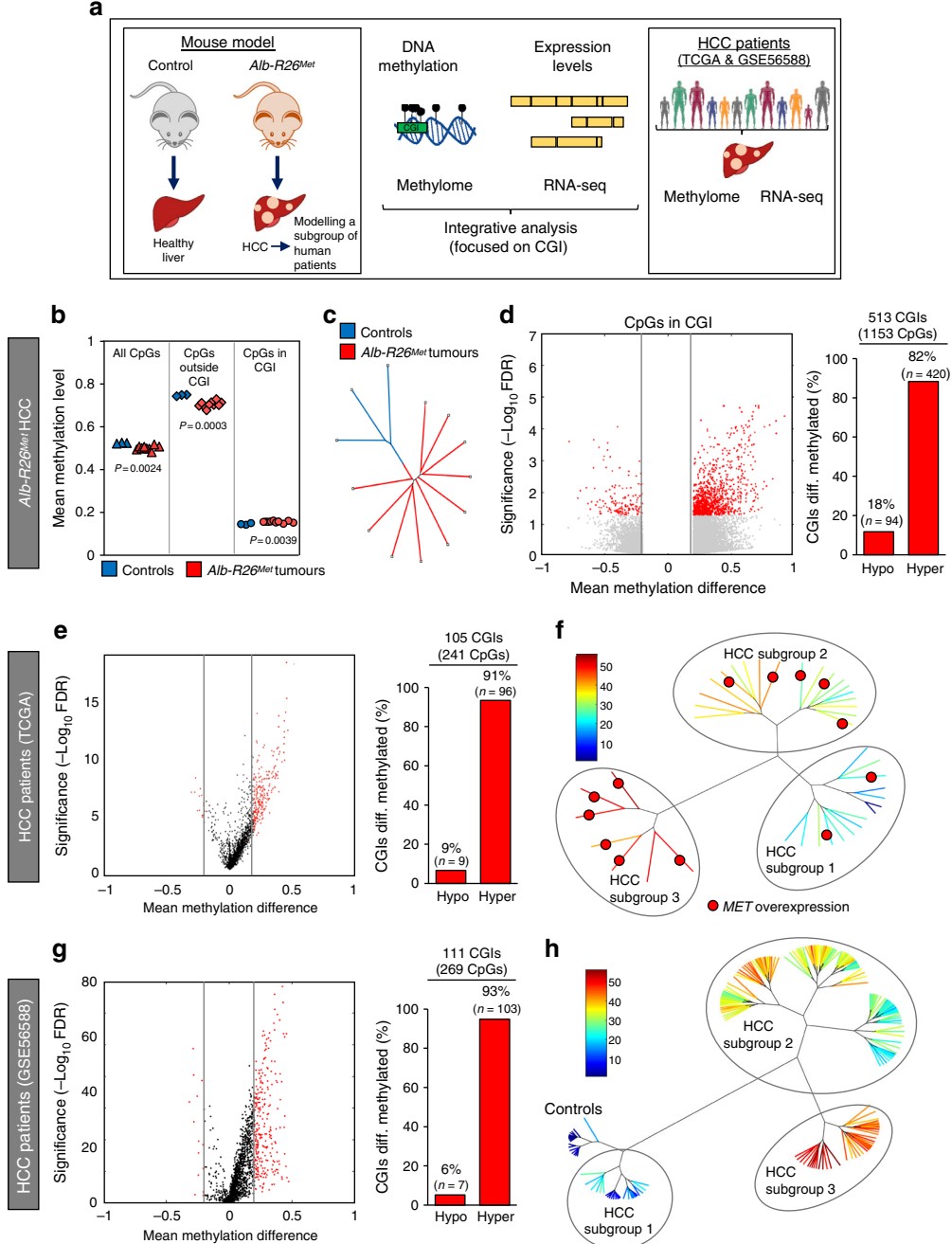

**Fig. 1** Methylome studies identify an enrichment of CGI hypermethylation in *Alb-R26^Met* tumours, also present in a subset of HCC patients. **a** Schematic representation of the overall strategy employed. DNA methylation and gene expression levels were analysed in *Alb-R26^Met* tumours and control livers. Outcomes were compared with HCC human database. **b** Mean methylation levels in controls and *Alb-R26^Met* tumours, focusing on all CpGs, CpGs outside CGIs and CpGs in CGIs. **c** Unrooted distance tree using the overall DNA methylation content subdivides *Alb-R26^Met* tumours and controls in two distinct clusters. **d** Volcano plot reporting methylation differences with significance (expressed as $-\text{Log}_{10}$ FDR) for CpGs in CGIs in *Alb-R26^Met* tumours versus control (left). Significant differences (methylation difference > 0.2 and FDR < 0.05) are shown in red. Graph reporting the percentage (and numbers) of hypomethylated versus hypermethylated CGIs (right). **e** Volcano plot reporting the mean methylation differences with significance (expressed as $-\text{Log}_{10}$ FDR) in HCC patients from TCGA (left) for differentially methylated CGIs identified in *Alb-R26^Met* tumours. Similar methylation levels in HCC patients and controls are reported in black, whereas changes (>0.2) are reported in red. On the right, the graph reports the percentage (and numbers) of hypomethylated versus hypermethylated CGIs. **f** Unrooted distance tree of the 41 TCGA HCC patients showing patient segregation in three distinct subgroups, according to the 416 CGIs found differentially methylated in *Alb-R26^Met* tumours. Red dots highlight patients in which *MET* is overexpressed ($\log_2$FC > 1). Patients are reported in different colours according to the percentage of overlap (the scale in percentage is shown on the left). Note the striking correlation between differentially methylated CGIs and *MET* overexpression in the HCC patient subgroup 3. **g, h** Volcano plot (**g**) and unrooted distance tree (**h**) from studies using a second cohort of 224 HCC patients and 10 controls (GSE56588 dataset)

tumour and adjacent liver as control (from The Cancer Genomic Atlas; TCGA[25]). For comparisons between mouse and human data, we first mapped the 513 identified mouse CGIs (mm9) to the corresponding CGIs in human (hg19), using the UCSC toolbox. 501 out of 513 CGIs were successfully matched between the two genomes. For 416 CGIs, human methylome data were available in TCGA dataset (Supplementary Fig. 3A; Supplementary Data 2). We extracted the methylation β-value for the CpGs within these human CGIs and calculated the mean methylation difference for each CpG and for each of the 41 HCC patients. While the majority of these CpGs showed a methylation difference below 0.2, a proportion of CGIs (24%) were differentially methylated (FDR < 0.05) with a hypermethylation enrichment score similar to the $Alb\text{-}R26^{Met}$ HCC (91%; Fig. 1e, Supplementary Fig. 3B). As the analysed cohort includes patients with widely diverse aetiologies and characteristics, we next analysed the 416 CGIs in the individual patients. Intriguingly, hierarchical clustering analysis segregated these HCC patients into three distinct subgroups, with one subgroup composed of seven patients reaching 43–56% overlap with the $Alb\text{-}R26^{Met}$ list (subgroup-3; Fig. 1f, Supplementary Fig. 3C). The relevance of the $Alb\text{-}R26^{Met}$ methylation changes in the context of human HCC was further assessed in a second distinct cohort of 234 human samples (224 HCC patients and 10 control individuals[26]). 27% of CGIs differentially methylated in $Alb\text{-}R26^{Met}$ HCC are also altered in human HCCs, again with an enrichment in hypermethylation (93%; Fig. 1g, Supplementary Fig. 4A). Moreover, these methylation changes distinguished controls from HCC patients, which further segregate into three subgroups. HCC subgroup-3 reaches about 50% CGI overlap with the $Alb\text{-}R26^{Met}$ list (Fig. 1h, Supplementary Fig. 4B).

Next, we asked whether there would be any correlation between MET alterations with the three human HCC subgroups identified by the $Alb\text{-}R26^{Met}$ methylome screening. Concerning the HCC patient cohort from TCGA, we were able to perform correlative studies as RNA-seq and mutation data are available. In particular, we analysed MET mutations and MET expression levels for each patient belonging to the 3 different HCC subgroups. All HCC patients carry the wild-type form of MET, which is in agreement with rare mutations of MET in HCC. Concerning expression levels, MET is overexpressed in 86% (6/7) of HCC patients belonging to subgroup-3 (which best overlaps with CGI methylation changes in $Alb\text{-}R26^{Met}$), in 32% (6/19) to HCC subgroup-2, and only in 13% (2/15) to HCC subgroup-1 (Fig. 1f; patients with MET overexpression are highlighted with a red dot; Supplementary Fig. 3C–F). For the HCC patient cohort from GSE56588, expression data (array) are only available for some patients and without information about mutations. Therefore, correlative studies were not possible with this HCC cohort. Together, these findings show that liver cancer modelled by the $Alb\text{-}R26^{Met}$ genetic system is characterised by methylation changes of specific CGIs, with a predominant hypermethylation profile. A high proportion of these alterations are also found in HCC patient subgroups. Furthermore, there is a striking correlation between differentially methylated CGIs and MET overexpression in the HCC patient subgroup modelled by the $Alb\text{-}R26^{Met}$ genetic setting.

**Enrichment in CGI hypermethylation is necessary for $Alb\text{-}R26^{Met}$ tumorigenesis**. The overall enrichment in CGI hypermethylation in the $Alb\text{-}R26^{Met}$ genetic system prompted us to determine its relevance for cell tumorigenic properties. We designed different demethylating experimental conditions using low doses of Decitabine (0.3 μM; Fig. 2a), according to previously reported protocols[12]. We used three different $Alb\text{-}R26^{Met}$ HCC

cell lines, established from individual $Alb\text{-}R26^{Met}$ tumours[24]. Decitabine treatment does not affect cell viability of $Alb\text{-}R26^{Met}$ HCC cells, as well as of MLP-29 cells, a mouse liver progenitor cell line that is not tumorigenic as illustrated by its inability to form colonies in anchorage-independent growth assays (Fig. 2b). Instead, Decitabine treatment interferes with $Alb\text{-}R26^{Met}$ cell tumorigenic properties, irrespective of the HCC cell line used, as exemplified by: (a) reduced colony numbers when cells are grown in an anchorage-independent manner (Fig. 2c); (b) reduced number and size of foci when cells are grown in an anchorage-dependent manner (Fig. 2d); (c) reduced tumour spheres when cells are grown in self-renewal conditions (Fig. 2e). The effect of global demethylation on cell tumorigenicity was further analysed in vivo by performing xenografts in nude mice. The tumour volume was significantly reduced in mice either injected with Decitabine pre-treated $Alb\text{-}R26^{Met}$ HCC cells or when Decitabine pulses were administered to mice injected with untreated $Alb\text{-}R26^{Met}$ HCC cells (Fig. 2f–left). Decitabine doses used in vivo were not toxic, as revealed by no significant changes on the mouse weight during the treatment (Fig. 2f–right). Together, these results indicate that the overall enrichment in CGI hypermethylation is functionally relevant for tumorigenesis modelled by the $Alb\text{-}R26^{Met}$ genetic system.

**CGI hypermethylation correlates with gene upregulation in $Alb\text{-}R26^{Met}$ HCC**. Alterations in DNA methylation are known to impact gene expression. We analysed the expression levels of the 431 genes with differentially methylated CGIs in $Alb\text{-}R26^{Met}$ tumours using high-coverage RNA-seq data (4 $Alb\text{-}R26^{Met}$ tumours and 4 control livers). Studies highlighted 93 genes differentially expressed ($\log_2 FC > 1$, FDR < 0.05; Supplementary Data 3). According to the Kyoto Encyclopaedia of Genes and Genomes (KEGG) database, several cancer-related pathways were significantly enriched, such as MAPK signalling, viral carcinogenesis, pathways in cancer, TGF-β signalling, cell cycle, renal cell carcinoma (Fig. 3a, Supplementary Fig. 5), strengthening the significance of genes differentially methylated and expressed in the $Alb\text{-}R26^{Met}$ cancer model. Remarkably, the top-ranked MAPK pathway is coherent with its essential functionality for $Alb\text{-}R26^{Met}$ tumorigenicity, as previously reported[24]. Among genes differentially methylated and expressed, 36 genes showed the expected inverse correlation between methylation and expression where 20 genes are hypomethylated and overexpressed, and 16 genes are hypermethylation and downregulated (Fig. 3b, Supplementary Data 3). Unexpectedly, 55 genes (59%) were found hypermethylated and overexpressed (Fig. 3b). Thus, tumorigenesis modelled by the $Alb\text{-}R26^{Met}$ mice is characterised by a set of genes with changes in CGI methylation accompanied by a reprograming of transcript levels.

The intriguing enrichment in hypermethylated and overexpressed genes drove us to analyse the position of the hypermethylated CGIs with respect to the ATG. Interestingly, the CGI of overexpressed genes is either close to the ATG or in the gene body region, in contrast to the CGI position of downregulated genes exclusively located around the ATG (Supplementary Fig. 6). Concerning the 55 genes hypermethylated and overexpressed, they can be subdivided into two groups. Group-I includes 31 genes, for which the CGIs are located between −50% and 30% relative to the ATG (predominantly into the 5′-UTR). Group-II includes 24 genes, whose CGIs are located much further from the ATG (from 30% of the gene body relative to the ATG to the transcription termination site), corresponding to gene body regions (Fig. 3c, Supplementary Fig. S6). Next, we analysed whether the CGI location influences gene expression. Intriguingly, the extent of overexpression is significantly higher

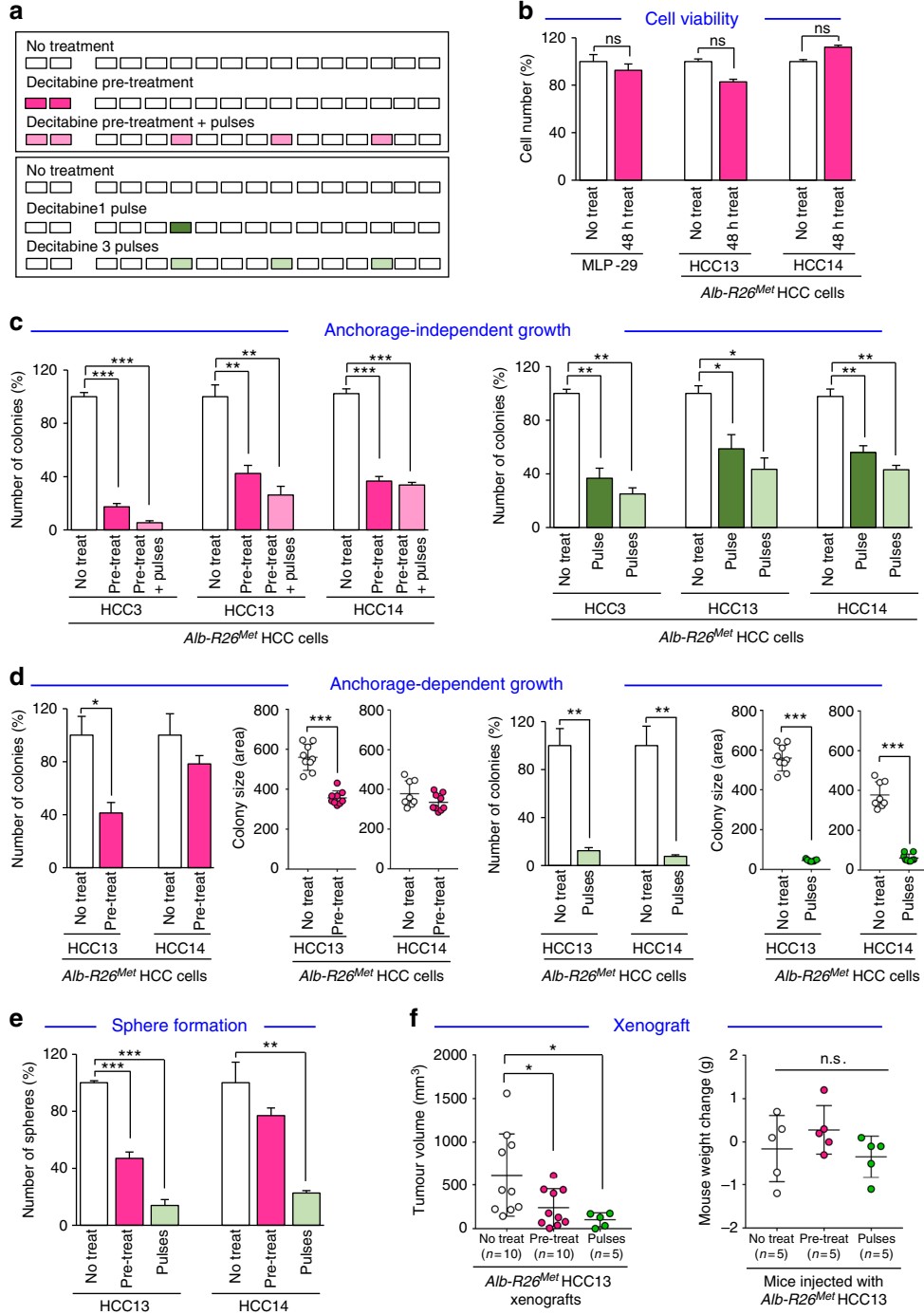

**Fig. 2** Global CGI hypermethylation is functionally relevant for *Alb-R26*$^{Met}$ tumorigenesis. **a** Scheme reporting demethylating treatments (Decitabine; 0.3 μM) used for in vitro and in vivo experiments with *Alb-R26*$^{Met}$ HCC cells. Cells were pre-treated (48 h) before using them for experiments with or without subsequent Decitabine pulses every 3 days (top; pink). Untreated cells were exposed to pulses of demethylating treatments during the assay (bottom; green). **b** Effect of Decitabine treatment (48 h) on cell viability of mouse liver progenitor MLP-29 cells and of *Alb-R26*$^{Met}$ HCC cells (HCC13 and HCC14). **c** Anchorage-independent growth (soft agar) assay using 3 different *Alb-R26*$^{Met}$ HCC cell lines (HCC3, 13 and 14) showing effects of demethylating treatments described in **a**. **d** Effect of Decitabine treatment in anchorage-dependent growth (foci formation) assay of 2 different *Alb-R26*$^{Met}$ HCC cell lines (HCC13 and 14). Graphs report number and size of colonies. **e** Decitabine (pre-treatment and treatment) reduces numbers of tumour spheres derived from *Alb-R26*$^{Met}$ HCC cells (HCC13 and HCC14). **f** Left: Graph reporting the tumour volume of mice injected either with untreated cells (No treat), with Decitabine pre-treated *Alb-R26*$^{Met}$ HCC cells (Pre-treat), or with untreated cells following pulses of Decitabine in vivo treatments (Pulses). Note that tumour volume was significantly reduced in mice injected either with Decitabine pre-treated *Alb-R26*$^{Met}$ HCC cells or with untreated cells following in vivo Decitabine pulses. Right: Mouse weight of the indicated groups was measured before and after xenograft experiments. No significant changes were observed, indicating that the dose of Decitabine used in vivo was not toxic. Significant differences between groups are indicated on the top. Not significant (ns): $P > 0.05$, $*P < 0.05$, $**P < 0.01$; $***P < 0.001$

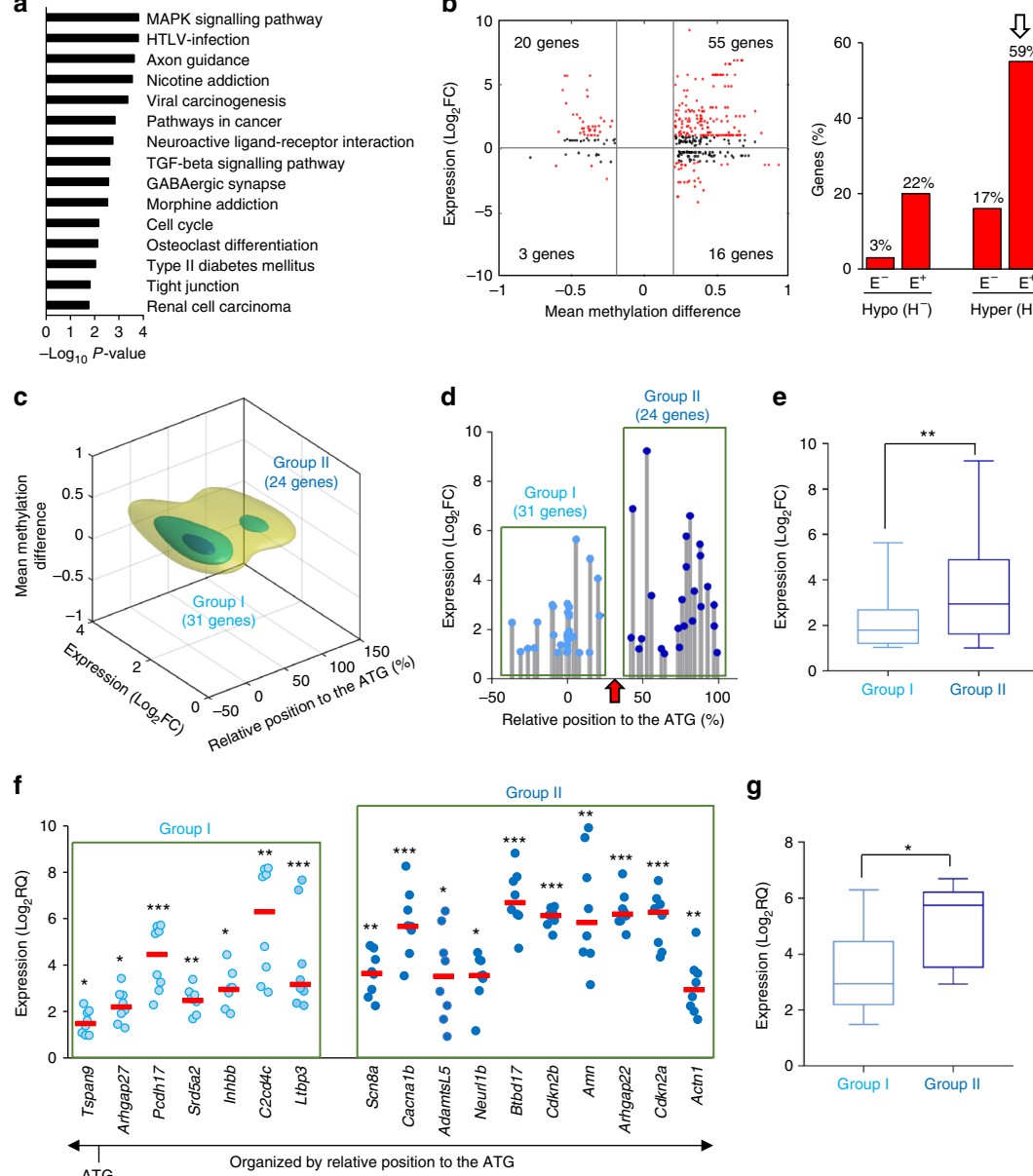

**Fig. 3** *Alb-R26^Met* tumours are characterised by an enrichment in genes overexpressed and with hypermethylated CGIs. **a** Histogram reporting the KEGG pathway enrichment analysis for genes with changes in CGI methylation and expression in *Alb-R26^Met* tumours, ordered according to the −Log$_{10}$ *P*-value. **b** Left: Methylation differences versus expression for all genes with CGIs hypermethylated (H$^+$) or hypomethylated (H$^-$) in *Alb-R26^Met* tumours. Expression values are relative to controls. Dots correspond to single differentially methylated CpG and the corresponding gene expression (genes which expression is significantly below or above Log$_2$FC ± 1 are indicated in red). Right: Graph reporting the percentage of downregulated (E$^-$) and upregulated (E$^+$) genes among those with a hypomethylated (H$^-$) or hypermethylated (H$^+$) CGI. Note the enrichment of genes overexpressed and with hypermethylated CGIs (indicated by an arrow), on which subsequent studies were focused. **c** For the 55 genes overexpressed and with hypermethylated CGI in *Alb-R26^Met* tumours, 3D density plot shows their distribution according to relative position to the ATG (as percentage), gene expression level (as Log$_2$FC) and CGI methylation (as β-value difference). Note that genes segregate into two groups, according to their relative position to the ATG. **d** Graph reporting the individual expression level (as Log$_2$FC; from RNA-seq data) of hypermethylated and overexpressed genes found in *Alb-R26^Met* tumours compare to controls. Note that the relative position of the hypermethylated CGI to the ATG well segregate the two groups (indicated with a red arrow). **e** Box plot illustrating the global expression levels of genes in Group-I and Group-II. **f** Graph reporting individual expression levels (as Log$_2$FC; by RT-qPCR) of genes belonging to the two groups in *Alb-R26^Met* tumours (*n* = 8) relative to control livers (*n* = 6). Red lines report the median Log$_2$FC in expression. Note that genes are distributed according to the location of the hypermethylated CGI relative to the ATG. **g** Box plot showing the global Log$_2$FC in expression (according to data in F) of genes in Group-I and Group-II. In **e** and **g**, the median is reported by a line and bars extend to the minimum/maximum values. Significance is indicated on the top. *P < 0.05, **P < 0.01, ***P < 0.001

for Group-II than Group-I (median $\log_2$FC = 2.15 ± versus median $\log_2$FC = 3.38 ± 0.4; Fig. 3d, e). Importantly, the promoter CGI methylation status of genes belonging to Group-II was similar in $Alb$-$R26^{Met}$ tumours and control livers, thus excluding that changes in promoter methylation could influence gene expression levels (Supplementary Data 4). We corroborated these results through RT-qPCR analysis of a subset of genes belonging to both groups in $Alb$-$R26^{Met}$ tumours ($n = 8$) relative to control livers ($n = 6$). Results showed consistent upregulation of all genes (Fig. 3f), with significant higher expression levels for those within Group-II (Fig. 3g). Together, these results highlight a set of overexpressed genes with hypermethylated CGIs in $Alb$-$R26^{Met}$ tumours and identify a correlation between the location of hypermethylated CGIs and transcription status, where CGIs located further from the ATG showing predominantly increased transcription.

Next, we analysed whether Decitabine treatment would affect in vivo the expression levels of genes found hypermethylated and overexpressed in the $Alb$-$R26^{Met}$ tumours. Focusing on a set of genes, we examined both their expression levels and the methylation levels of their corresponding CGIs in dissected tumours from $Alb$-$R26^{Met}$ mice either untreated or treated with Decitabine. RT-qPCR results showed that Decitabine treatment significantly decreases the expression levels of 7/8 analysed genes (Fig. 4). Bisulfite sequencing studies revealed decreased methylation levels of most CpGs within the gene body CGIs (Fig. 4, Supplementary Data 5). Thus, CGI hypermethylation of genes belonging to Group-I and Group-II ensures their increased expression levels in $Alb$-$R26^{Met}$ tumours as demethylating treatment leads to a reduction of both CGI methylation content and transcription.

**A CGI hypermethylation and gene overexpression signature defines a HCC patient subset.** Next, we explored the relationship between changes in CGI methylation and gene expression in the above cohort of 41 HCC patients. Because of expected epigenomic heterogeneity between human samples, we reasoned it relevant to perform analyses in individual patients. We integrated transcriptome and methylome data to extract the expression levels of genes with differentially methylated CGIs (Fig. 5a), then classified patients according to the highest percentage of genes: (a) overexpressed with hypermethylated CpGs (H$^+$E$^+$); (b) overexpressed with hypomethylated CpGs (H$^-$E$^+$); (c) underexpressed with hypermethylated CpGs (H$^+$E$^-$); (d) underexpressed with hypomethylated CpGs (H$^-$E$^-$). Intriguingly, 23/41 patients (56%) showed an enrichment of genes overexpressed and with hypermethylated CpGs (H$^+$E$^+$ patient-subset; Fig. 5a, Supplementary Data 6), similar to the $Alb$-$R26^{Met}$ model (Fig. 3b). Analysis of $MET$ levels in HCC patients revealed that the mean $MET$ levels in the H$^+$E$^+$ subset is 0,77 ± 0,16 (9/23; 39% patients with $MET$ levels ≥ 1), whereas in the "NO H$^+$E$^+$" subset is 0,2 ± 0,24 (5/18; 27% patients with $MET$ levels ≥ 1; Supplementary Data 6). Interestingly, all 7 patients belonging to the HCC subgroup-3 (in Fig. 1f) are characterised by more than 37% of genes both hypermethylated and overexpressed, and 5/7 patients belong to the H$^+$E$^+$ subset (these 7 patients are highlighted with a red square and red % in Fig. 5a). Next, we asked whether the H$^+$E$^+$ patient subset could be also identified according to global gene expression or methylation features. Unsupervised cluster analysis of either global methylome or transcriptome data did not lead to the same patient clustering (Supplementary Fig. 7), thus strengthening the usefulness of combining methylation-expression features to identify specific HCC patient subsets.

The remarkable correlation between data obtained in the $Alb$-$R26^{Met}$ HCC model and analysed patient samples prompted us to perform integrative studies using another HCC model, for which methylation and expression data are available: the hepatitis-B virus-X mice ($HBx^{tg}$; GSE48052[27]). We first identified all CpGs differentially methylated in $HBx^{tg}$ HCC model, then correlated them with gene expression levels. We identified 115 genes both differentially methylated and differentially expressed (a very similar number to the 97 genes found in the $Alb$-$R26^{Met}$ genetic setting). Nevertheless, we found a different distribution compared to that of the $Alb$-$R26^{Met}$ HCC, with an enrichment in genes both hypomethylated and downregulated (Supplementary Fig. 8). Next, we performed correlative analyses with the 41 HCC patients (reported in Fig. 5a): amongst the 18 "NO H$^+$E$^+$" subset, 8 patients (20%) share the same enrichment of hypomethylated and downregulated genes modelled by the $HBx^{tg}$ mice. Unexpectedly, only 1/8 of these patients is reported positive for HBV. Thus, an epigenetic rewiring of gene sets through hypomethylation and downregulation occurs in a fraction of HCC patients, who do not appear to be characterised by the HBV-associated risk. Collectively, these findings indicate a rather intriguing specificity in how genes are epigenetically reprogrammed in HCC patients: an enrichment in hypermethylated and upregulated genes (for those corresponding to the $Alb$-$R26^{Met}$ model) versus an enrichment in hypomethylated and downregulated genes (for those corresponding to the $HBx^{tg}$ model).

For the several genes found overexpressed and with hypermethylated CGIs in the H$^+$E$^+$ patient subset, such as $WT1$, $DLK1$, $TP73$, $EEF1A2$, $IGF1R$, $DKK1$, $SPOCK1$, $ITPKA$, $HOXA3$, $NOX4$, $FZD10$, $VASH2$, $GATA2$, $SOX8$, their upregulation in HCC samples is coherent with their reported function as oncogenes in cancer. Concerning the H$^+$E$^+$ patient subset, based on clinical data from TCGA, no association was found with a specific risk factor, such as HBV/HCV infection, high-alcohol intake or non-alcoholic fatty liver disease (NAFLD) (Supplementary Fig. 9). Instead, the H$^+$E$^+$ patient subset is distinguished by specific HCC molecular markers[28]. In particular, analysis of available RNA-seq data revealed a significant upregulation of $alpha$-$FETOPROTEIN$ ($AFP$; a HCC marker when expressed in adult livers), $JAG1$, $NOTCH3$, $NOTCH4$, $SOX9$, $VIM$ (progenitor markers) and $CD24$ (a HCC prognosis marker; Fig. 5b). Importantly, these markers are also upregulated in $Alb$-$R26^{Met}$ HCC (Fig. 5c), as we recently reported[24]. Together, these results show that an enrichment in genes characterised by "CGI hypermethylation and overexpression" occurs in HCC patients belonging to the so-called "proliferative-progenitor" subclass[28]. Moreover, these HCC patients share common features with the $Alb$-$R26^{Met}$ liver cancer model: the epigenetic H$^+$E$^+$ signature and the "proliferative-progenitor" cell feature.

**Overexpressed genes with hypermethylated CGIs in 5'-UTR or gene body regions act as oncogenes.** The intriguing overlap between the $Alb$-$R26^{Met}$ model and the H$^+$E$^+$ patient subset prompted us to explore the relevance in cancer of the 55 genes found in $Alb$-$R26^{Met}$ tumours both overexpressed and with hypermethylated CGIs either in the 5'-UTR or in the gene body region. For this analysis, transcriptome data from HCC patients were available for 51/55 genes (Supplementary Data 7). Remarkably, most genes are overexpressed in a large proportion of HCC patients (Fig. 6a, Supplementary Data 8), with a significant higher number in the H$^+$E$^+$ patient subset compared with the other (Fig. 6b). These genes include $PRRX1$ (28 patients out of 41), $CLDN7$ (20), $DBN1$ (25), $PCDH17$ (30), $PTK7$ (21), $ADAMTSL5$ (21), $ARHGAP21$ (30), $NFKB2$ (23), $CDKN2B$ (30), $RELB$ (22), $DUSP8$ (24), $SSBP4$ (20), $IRX3$ (27), $NEURL1B$ (19). Differences in the expression of these 51 genes permitted segregating the H$^+$E$^+$ patient subset from the other (Fig. 6c).

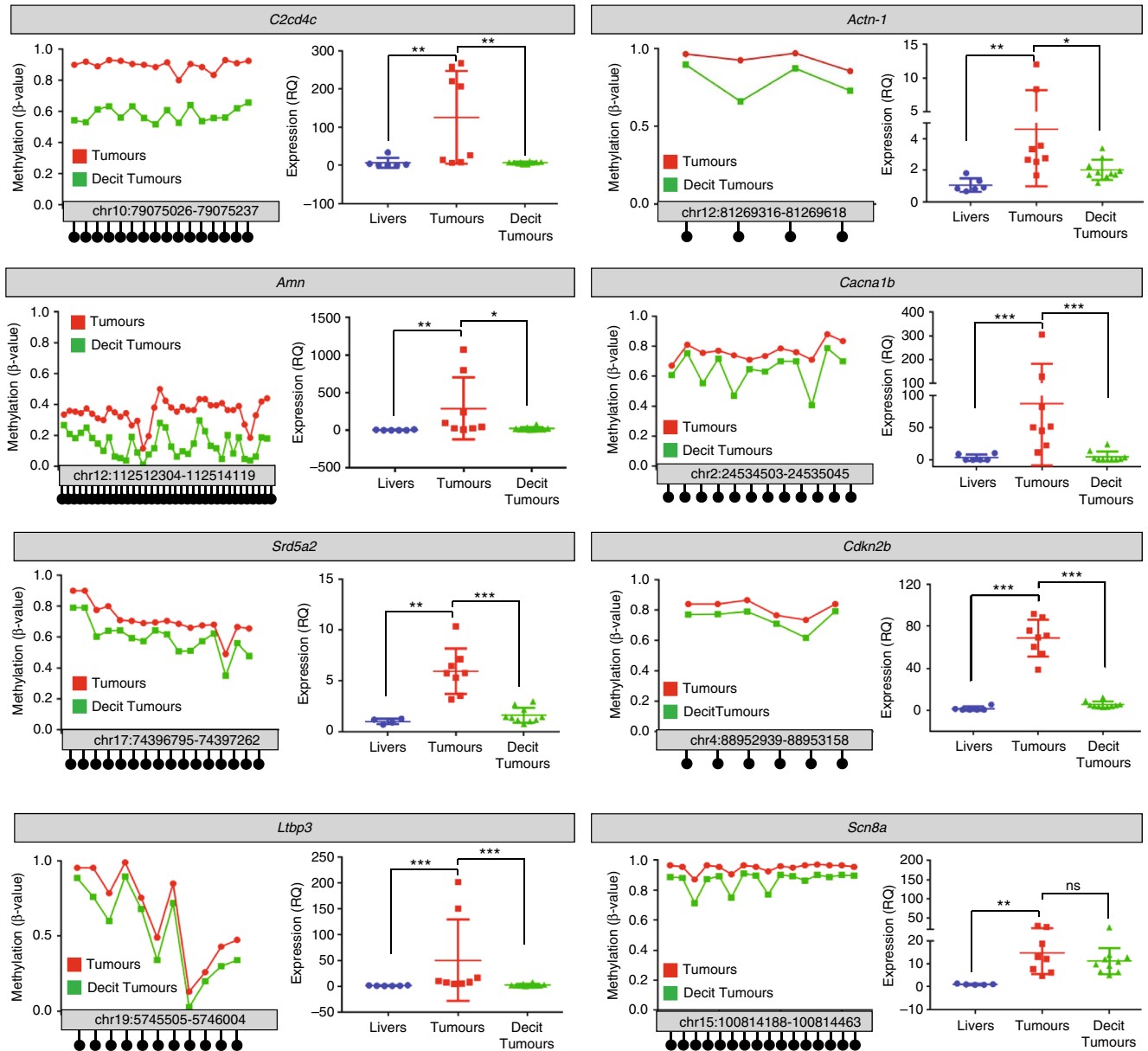

**Fig. 4** Decitabine treatment decreases the expression and the CGI methylation levels of genes hypermethylated and overexpressed in *Alb-R26^Met* tumours. Expression and CGI methylation levels of a set of genes found hypermethylated and overexpressed in the *Alb-R26^Met* tumours were analysed in dissected tumours from *Alb-R26^Met* mice either untreated (red) or treated with Decitabine (green). For each indicated gene, graphs report the methylation levels of CpGs within the CGI of interest (left) and the expression levels of genes (right) in tumours from *Alb-R26^Met* mice either untreated (red) or treated with Decitabine (green), compared to control livers (blue). Note that demethylating treatment significantly decreased transcription levels. Concerning the *Scn8a* gene, the methylation levels of its gene body CGI was reduced in Decitabine treated tumours compared to untreated tumours. This was accompanied by a trend in downregulation of its expression levels, although not significant. It is possible that for *Scn8a*, the demethylation extent caused by the dose of Decitabine used is suboptimal to significantly influence its expression levels. Alternatively, a more complex mechanism could be involved in the regulation of *Scn8a* expression. Significance is indicated on the top. Not significant (ns): $P > 0.05$, *$P < 0.05$, **$P < 0.01$, ***$P < 0.001$

Furthermore, for each HCC patient we analysed the methylation levels of the CGIs corresponding to the 55 genes. We took into account that the number of CGIs for each gene varies between genes (Supplementary Data 7). 53/55 genes successfully lifted-over from mouse to human, and both methylation and expression data are available for 51 genes. These analyses revealed that 42/51 (82%) genes are both hypermethylated and overexpressed in at least 1 patient, and that 40/41 (97,5%) patients have at least 1 gene both hypermethylated and overexpressed (Fig. 6d, e, Supplementary Fig. 10, Supplementary Data 9). Additionally, there is a significant higher number of genes both hypermethylated and

overexpressed in the H+E+ patient subset compared to the "NO H+E+" subset (H+E+ versus "NO H+E+": *P*-value < 0.001; Supplementary Fig. 10).

Curiously, in the *Alb-R26^Met* cancer model *Cdkn2a*, rather considered as a tumour suppressor, is overexpressed and hypermethylated in its gene body CGI, whereas no methylation changes were observed in its promoter CGI (Supplementary Data 4). We examined whether this phenomenon would also occur in HCC patients by analysing *CDKN2A* methylation and expression in HCC patients from TCGA and GSE56588 cohorts (for which methylation and expression data are available: 205/224 patients).

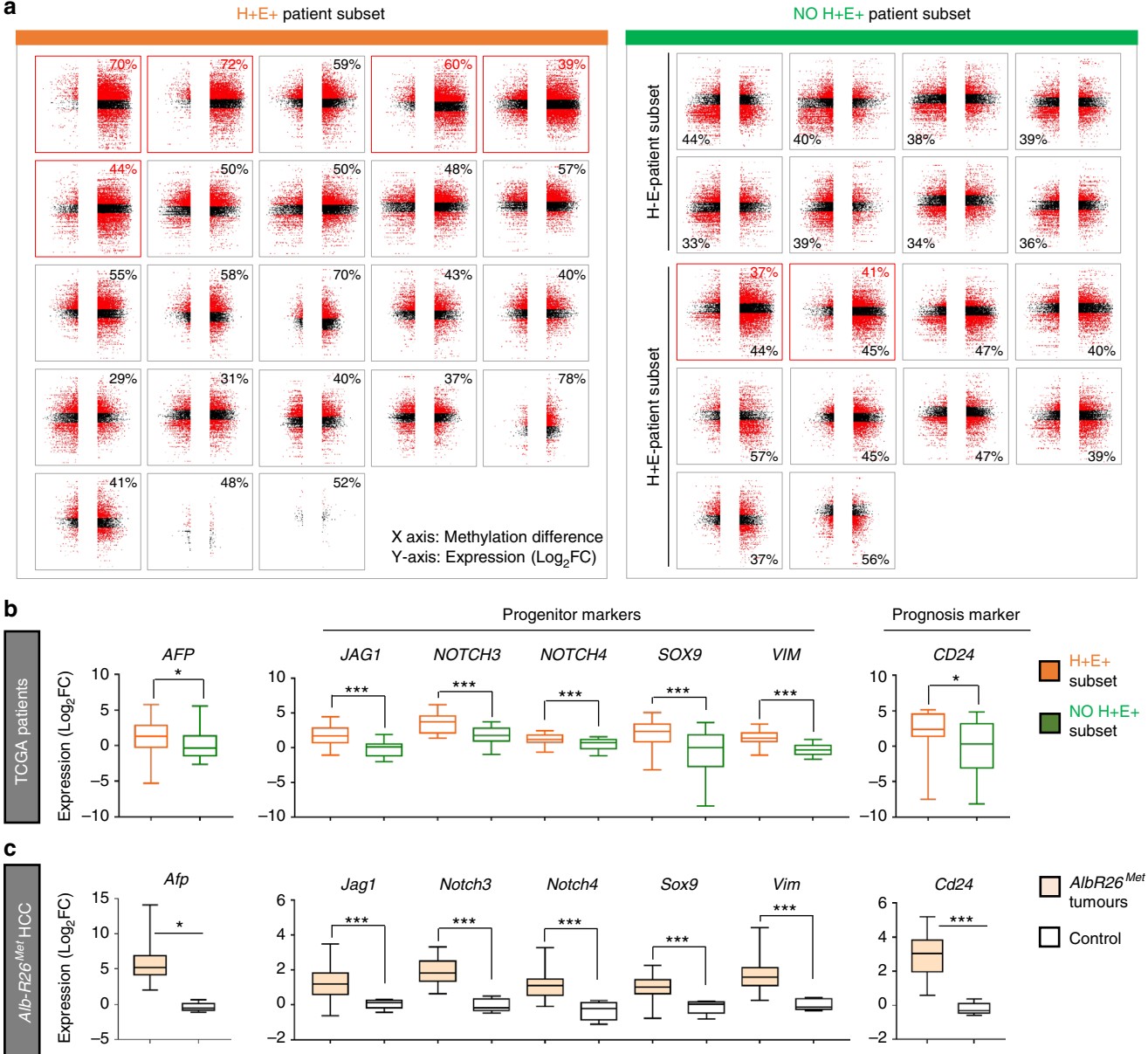

**Fig. 5** A HCC patient subset, which is characterised by an enrichment of genes overexpressed and with hypermethylated CGIs, belongs to the HCC "proliferative-progenitor" subclass. **a** The 41 HCC patients are classified according to the highest percentage of genes over- versus underexpressed and with hyper- versus hypomethylated CGIs. In orange (left), patients with an enrichment of genes overexpressed and with CGI hypermethylation (H+E+ patient subset). Patients are organised according to the absolute number of hypermethylated CGIs. The percentage of genes overexpressed and hypermethylated is reported on the top. In green (right), all other patients are reported (NO H+E+ patient subset). Note that this patient subset is characterised by an enrichment in downregulated genes. Patients are organised according to an enrichment of genes with CGI hypomethylation (top) and hypermethylation (bottom). Concerning the 7 patients of the HCC subgroup 3 identified in Fig. 1f (corresponding to the best overlap patients), 5 of them belong to the H+E+ subset. Notably, all of these 7 patients are characterised by more than 37% of genes both hypermethylated and overexpressed (highlighted in panel with a red square and a red percentage of genes overexpressed with hypermethylated CGI). The X-axis reports methylation differences, whereas the Y-axis reports expression as Log2FC. **b** Transcript levels (from RNA-seq data) of the indicated genes in H+E+ patient subset (in orange) versus the others (in green). Note significant high transcript levels of *AFP, JAG1, NOTCH3, NOTCH4, SOX9, VIM* and *CD24* in the H+E+ patient subset. **c** Transcript levels by RT-qPCR for the same genes shown in **b** analysed in *Alb-R26*[Met] tumours versus control livers, displaying the same profile of gene upregulation as in the H+E+ patient subset. Data have been reported in ref.[24]. Significance is indicated on the top. *P < 0.05, ***P < 0.001

Mouse *Cdkn2a* has two CGIs: one in the promoter and another in the gene body. Instead, human *CDKN2A* has 5 CGIs: one in the promoter and four in the gene body. Data are available only for the CGI in the promoter and for one of the four CGIs located in gene body. Notably, in both cohorts we found an enrichment of patients with an overexpression of *CDKN2A* (39/41 and 166/204, in the respective cohorts), which is associated to a

hypermethylation of the gene body CGI (21/39 and 163/166, in the respective cohorts). In contrast, not methylation changes have been detected in the promoter CGI for both HCC cohorts (Supplementary Fig. 11).

Analysing pathway enrichments in KEGG pathways of genes overexpressed with hypermethylated CGIs, we identified a significant enrichment of several cancer-related pathways, such

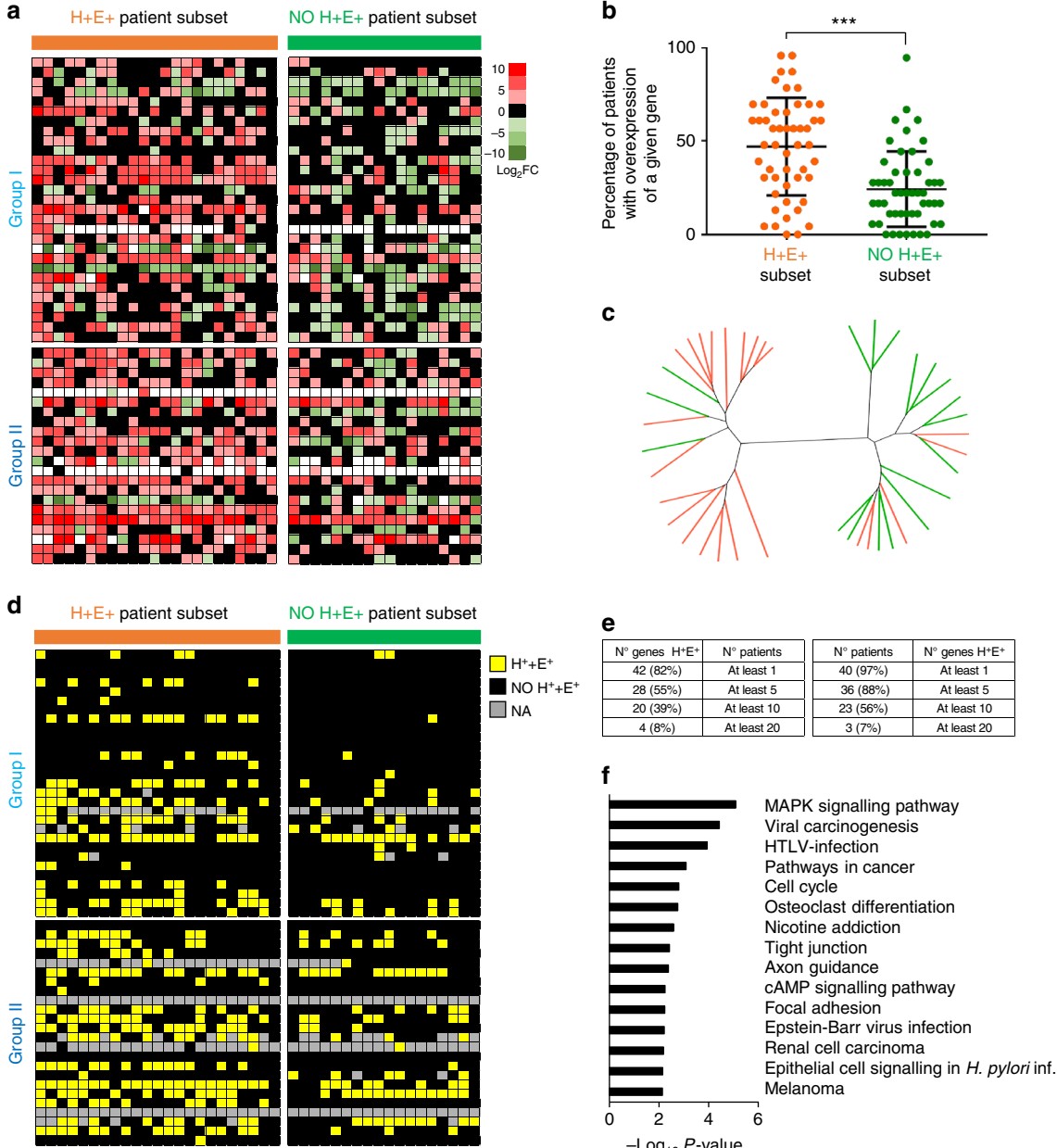

**Fig. 6** The 55 genes identified in *Alb-R26*$^{Met}$ tumours are also upregulated in the H$^+$E$^+$ patient subset, and a large proportion of them is characterised by hypermethylated CGIs. **a** Heat-map reporting expression levels of genes found overexpressed and with hypermethylated CGIs in *Alb-R26*$^{Met}$ tumours versus control livers (rows; subdivided in Group I and II and organised according to the relative position to the ATG) in individual HCC patients (columns; organised as in Fig. 5a). Red: upregulated genes; green: downregulated genes. Black: not differentially expressed. White: data not available. The scale is shown on the right (expressed as Log$_2$ FC). The H$^+$E$^+$ patient subset is highlighted in orange (left), whereas all other patients in green (right). **b** In the graph, each dot corresponds to a given gene (the total 51 genes are reported). Their position corresponds to the percentage of patients in which the gene is overexpressed. In orange (left), for the H$^+$E$^+$ patient subset. In green (right), for all other patients. **c** Unrooted distance tree of HCC patients based on the 51 genes identified in the *Alb-R26*$^{Met}$ tumours. In orange: H$^+$E$^+$ patient subset. In green: all other patients. **d** Heat-map highlighting with a yellow square genes overexpressed and with hypermethylated CGI in the corresponding HCC patient. The heat-map is organised as in panel A. Black square: genes not simultaneously overexpressed and with hypermethylated CGI in the corresponding HCC patient. Grey: data not available. **e** Tables reporting the numbers with percentages of genes overexpressed and with hypermethylated CGI in at least 1, 5, 10 or 20 HCC patients (left), as well as numbers with percentages of patients with at least 1, 5, 10 or 20 genes overexpressed and with hypermethylated CGI out of the 55 genes identified in the *Alb-R26*$^{Met}$ model (right). **f** Histogram reporting the KEGG pathway enrichment analysis for the 55 genes identified in *Alb-R26*$^{Met}$ tumours, ordered according to the −Log$_{10}$P-value. Significance is indicated on the top. ***$P < 0.001$

as MAPK signalling, viral carcinogenesis, pathways in cancer, cell cycle (Fig. 6f, Supplementary Fig. 12). Consistently, some of these genes are well-known oncogenes, such as *GRB10*, *MAP3K6*, *JUN* (which belong to MAPK pathway), *NFKB2*, *RELB* (which belong to NFkB and MAPK pathways), *MET*, *PTK7* (Supplementary

Data 10). The presence of poorly characterised genes among well-established oncogenes prompted us to explore their functional relevance in cell tumorigenic properties. Focusing on *Scn8a*, *Actn1*, *Srd5a*, *NFkB2* and *Neurl1b*, we used shRNA-mediated targeting to lower their expression levels in *Alb-R26*$^{Met}$ HCC cells

(Fig. 7a, Supplementary Fig. 13). Stable clones were used to assess cell tumorigenic properties in vitro and in vivo. These genes were selected because: (1) of their overexpression in HCC patients (*SCN8A* in 41%, *ACTN1* in 22%, *SRD5A2* in 5%, *NFkB2* in 56%, *NEURL1B* in 46%); (2) of their action as oncogenes in cancer cells (and particularly in HCC) has been less explored in previous studies (with the exception of *NFkB2* and *SRD5A*). Downregulation of either *Scn8a*, *Actn1*, *Srd5a*, *NFkB2* or *Neurl1b* interferes with the capability of cells to form: (a) colonies in anchorage-independent assays (Fig. 7b); (b) foci in anchorage-dependent assays, as revealed by a significant smaller foci size even if numbers were similar (Fig. 7c); (c) tumour spheres when cells were grown in self-renewal conditions (Fig. 7d); (d) tumours in nude mice xenografts (Fig. 7e). Collectively, these data show that most of the 55 genes identified in the *Alb-R26^Met* cancer model are also overexpressed and with hypermethylated CGIs in a large proportion of HCC patients, with a set of them acting together as an "oncogenic module".

## Discussion

The increasing knowledge on how epigenetic modifications such as DNA methylation influence patterns of gene expression in cancer holds great promises for understanding biological processes, as well as for their use in prognosis, patient stratifications and therapeutic intervention[3,29]. This is well exemplified by reports showing correlations between changes in CGI methylation and a remarkable resetting of transcriptional networks in cancer. In the present study, we employed a clinically relevant cancer mouse model in which tumorigenesis is triggered by a slight perturbation in signalling dosages rather than drastic genetic modifications, to examine the DNA methylation landscape associated with tumorigenic acquisition. We reasoned that such a genetic tool offers a unique way to model DNA methylation changes occurring in human cancerogenesis in the absence of drastic alterations of epigenetic modulators. Our genome-wide strategy highlighted key correlations between site-specific DNA methylation changes and transcriptional dosages of the corresponding genes. The type of changes found for some genes belong to the well-known mechanism of downregulation of tumour suppressors through promoter DNA hypermethylation, which was the case of *Oat* and *Igfbp5* that can act as tumour suppressors in certain cellular contexts. Quite unexpectedly, however, there is an enrichment of genes both overexpressed and with hypermethylated CGIs. Several of them are well-known oncogenes, such as *Grb10*, *Map3k6*, *Jun*, *RelB*, *Met*, *Ptk7*, as well as *NF-KB2*, *Srd5a2*, which have been functionally validated in this study together with others poorly investigated so far: *Scn8a*, *Actn1*, *Neurl1b*. Results from our functional assays in *Alb-R26^Met* HCC cells demonstrate how downregulating each individual oncogene reduces, but not abolishes, cell tumorigenic properties. These results conceptually illustrate that, although each oncogene contributes to the tumorigenic properties of cancer cells, they operate in a cooperative manner as an "oncogenic module" for ensuring robustness of the tumorigenic program.

Our integrative studies using human HCC databases demonstrate that enrichment in genes both overexpressed and with hypermethylated CGIs also characterises 56% of the HCC patients, which we named as the "H+E+ patient subset". For several genes, upregulation in expression levels is coherent with them being bona fide oncogenes. For example, it is the case of *WT1*, *DLK1*, *TP73*, *EEF1A2*, *IGF1R*, *DKK1*, *SPOCK1*, *ITPKA*, *HOXA3*, *NOX4*, *FZD10*, *VASH2*, *GATA2*, *SOX8*. Thus, our genetic studies together with a revisited analysis of human cancer databases reveal that raising dosages of oncogene sets characterised by hypermethylated CGIs is a robust mechanism

operating in cancer. The existence of such events in human pathology supports the clinical relevance of these findings. Remarkably, the H+E+ patient subset belongs to the "HCC proliferative-progenitor" subclass, thus attributing an additional feature to this aggressive HCC subtype. For clinical implementation, integrative methylome and transcriptome analyses on additional HCC cohorts will demonstrate the robustness of H+E+ patient subset classification. These findings also raise the question as to whether the H+E+ patient subset could be most sensitive to therapies based on demethylation agents[30].

Our expression analyses revealed that H+E+ genes can be segregated into two groups, according to their relative position to the ATG, with overall significant higher expression levels observed the further the CGI is located from the ATG (Group-II). Whereas for Group-I the relative position of the hypermethylated CGIs falls predominantly with the 5′-UTR, their location in Group-II is in the gene body. The positive correlation between gene body hypermethylation and expression is coherent with previous studies based on in vitro modulation of the methylation content in cancer cell lines[12]. Additionally, single-base resolution DNA methylation profiling combined with transcriptome analysis correlated changes in gene expression levels with the CpG methylation content in gene body[31–34]. The significance of gene body DNA methylation on transcriptional regulation is strengthened by studies exploring correlations with chromatin modifications. It has been reported that in the gene body: (a) H3K4me3 association to alternative promoters depends on their CpG methylation content, impacting alternative transcript products[35]; (b) H3K36me3 associates with methylated DNA in gene body and permits transcription[36]; (c) CTCF binding is lost in hypermethylated CGI, influencing splicing, in addition to the well-known action of CTCF in maintenance of chromatin architecture through generation of chromatin barriers[37,38]; (d) H3K27me3 and H3K9me3, known as repressive histone marks, are not associated with methylated DNA[39]. Future studies integrating methylome, transcriptome and ChIP-seq with several chromatin marks like those mentioned above will contribute to uncover the underlying mechanisms of action of oncogene upregulation through gene body methylation. Taking into account the variety of chromatin factors found associated in gene body, it is likely that different sets of genes are modulated by different mechanisms of action.

For translating these findings into therapies, an intriguing question is whether and to what extent the epigenetic reprogramming of a set of genes acting as an "oncogene module" still leaves space for tumour vulnerability. Our functional studies show that targeting each individual oncogene reduces, but not abolishes, tumorigenicity, indicating that each oncogene provides a net contribution to the whole tumorigenic properties of cancer cells. Such context may be particularly relevant for tumours that are not predominantly "addicted" to genetic mutation(s)[40], such as HCC. This likely explains the partial response of HCC patients even with most promising drugs targeting one or at least a restricted number of targets (e.g., Sorafenib). Such scenario contrasts with exceptional cases of effectiveness, due to the stringent addiction of cancer cells to a given oncogene, such as BCR-ABL in chronic myeloid leukaemia, ERBB2 in breast cancer, ERBB1 in non-small cell lung cancer, B-RAF in metastatic melanoma. To identify vulnerability, an approach could be to extract enriched pathways that are deregulated from the whole list of epigenetically reprogramed genes. In the case of tumorigenesis modelled by the *Alb-R26^Met* mice, the MAPK signalling cascade is on the top of the list of enriched pathways (with 11 genes differentially methylated in tumour versus control livers). Through a phosphokinome-based educated guess drug screen, we recently reported that tumorigenesis modelled by the *Alb-R26^Met* genetic

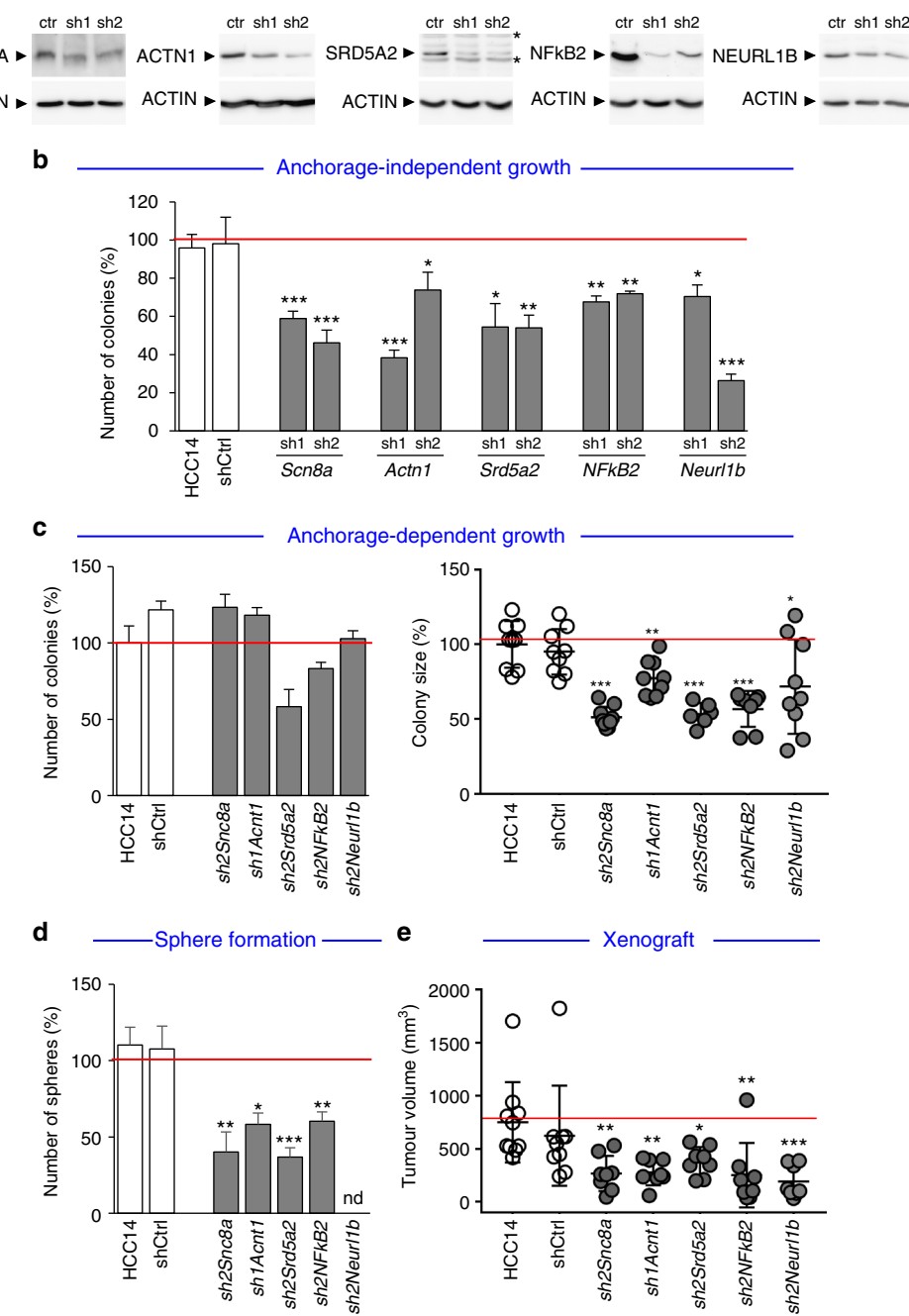

**Fig. 7** Downregulation of overexpressed genes with hypermethylated gene body CGI in *Alb-R26^Met* HCC cells interferes with their tumorigenic properties both in vitro and in vivo. **a** Western blots showing SCN8A, ACTN1, SRD5A2, NFkB2 and NEURL1B protein levels in stable clones established after transfection of *Alb-R26^Met* HCC14 cells with plasmids carrying a shRNA sequence targeting the corresponding gene. Protein levels were compared to control cells (ctr). ACTIN was used as a loading control in all western blots. The asterisk indicates nonspecific bands detected using anti-SRD5A2 antibodies. **b**–**e** Biological assays to assess functional properties of *Alb-R26^Met* HCC14 cells carrying a shRNA sequence targeting candidate genes. Effects were compared to HCC14 cells either untransfected or transfected with a control shRNA (shCtrl). **b** Graph reporting the number of colonies formed in anchorage-independent growth assays using 2 different shRNA targeting sequences for each candidate gene. Note a decrease in colony number formation of cells with downregulated candidate genes compared with control cells. **c** Graphs reporting the number (left) and the size (right) of colonies formed in anchorage-dependent growth assays. Whereas no significant changes in colony numbers were detected, note a significant decrease in colony size when the candidate gene is downregulated. **d** Graph reporting number of spheres formed in tumour sphere assays. Note that downregulation of candidate genes significantly reduces sphere number formation. **e** Graph reporting the tumour volume of mice injected either with *Alb-R26^Met* HCC14 control cells or with *Alb-R26^Met* HCC14 cells carrying a shRNA sequence targeting candidate genes. Note that downregulation of candidate genes significantly interferes with the in vivo tumorigenic properties of *Alb-R26^Met* HCC14 cells. Significant differences between groups are indicated on the top. *$P < 0.05$, **$P < 0.01$, ***$P < 0.001$ (nd: no determined)

system is vulnerable to Ras pathway targeting, provided that its inhibition occurs concomitantly while destabilising the stress support mitochondrial pathway[24]. It is therefore tempting to speculate that a proportion of tumours, particularly those with epigenetic reprogramming rather than those with drastic genomic instability, still maintains vulnerability to synthetic lethal interactions. Alternatively, the use of epigenetic modulating agents to reprogram a set of genes in cancer cells, ideally used at minimal doses to limit side effects[3,29], could reinforce the action of promising targeted therapies that, when used alone, have been unsatisfactory in clinical trials, as reported for chronic myeloid leukaemia[41]. We show here that Decitabine treatment reduces the methylation levels of gene body CGIs and the expression levels of the corresponding genes. Such event correlates with reduced tumorigenic properties of $Alb$-$R26^{Met}$ HCC cells. Nevertheless, cells for 10 days in culture after 48 h with Decitabine recover their tumorigenic properties (Supplementary Fig. 14), illustrating how reduced tumorigenicity by transient demethylation treatment is reversible, likely by resetting increased levels of oncogenes. This would be coherent with previous studies showing the capability of cancer cells to restore acquired epigenetic modifications[12]. Collectively, these findings support the possibility of achieving effective response in cancer combining epigenetic modulating agents with targeted treatments.

In conclusion, by exploring epigenetic changes associated with tumorigenesis in a clinically relevant mouse model, we discovered that for oncogene sets, characterised by hypermethylated CGIs either in their 5′-UTR or in the gene body, their expression levels are raised in cancer. The use of a mouse model in which tumorigenesis is not caused by drastic genetic manipulations strengthens the advantage of disrupting multiple oncogenes through an epigenetic reprogramming. Delineating the relationship between aberrant DNA methylation and expression of oncogenes/tumour suppressors will likely contribute to identify biomarkers for patient stratifications, functional pathways operating in cancer and strategies for an epigenetic restoration of deregulated genes in combination with molecular therapies.

## Methods

**Mice**. Ethics Statement: All procedures involving the use of animals were performed in accordance with the European Community Council Directive of 22 September 2010 on the protection of animals used for experimental purposes (2010/63/UE). The experimental protocols were carried out in compliance with institutional Ethical Committee guidelines for animal research (comité d'éthique pour l'expérimentation animale—Comité d'éthique de Marseille; agreement number D13-055-21 by the Direction départementale des services vétérinaires—Préfecture des Bouches du Rhône).

$Alb$-$R26^{Met}$ mice: $R26^{stopMet}$ and $Alb$-$R26^{Met}$ mice have been described previously[42,43]. Briefly, $R26^{stopMet}$ mice (international nomenclature $Gt(ROSA)$ $26Sor^{tm1(Actb-Met)Fmai}$) carrying a conditional mouse–human chimeric $Met$ transgene in the $Rosa26$ locus were crossed with $Albumin$-$Cre$ mice ($B6.Cg$-$Tg(Alb$-$cre)21Mgn/J$) obtained from the Jackson Laboratory. All mice were maintained on a 50% mixed 129/SV and C57BL6 background. Mice were genotyped via PCR analysis of genomic DNA as reported in previous studies[42,43]. Mice were housed under pathogen-free conditions.

Mice drug treatment: For in vivo demethylation experiments to asses methylation levels of selected CGIs, as well as expression levels of the corresponding genes, $Alb$-$R26^{Met}$ mice were treated with intraperitoneal injections of 2.5 mg/kg of Decitabine, twice per week (for a total of three injections).

**DNA/RNA-related experiments**. Genomic DNA isolation: Genomic DNA from $Alb$-$R26^{Met}$ tumours and control livers was prepared using the ZR Genomic DNA Tissue Miniprep (Zymo Research Company), according to the manufacturer's instructions.

Total RNA extraction: Total RNA from frozen tissues and cultured cells was isolated using the RNeasy Mini Kit (Qiagen), according to the manufacturer's instructions. DNase (Qiagen) treatment was included to avoid possible genomic DNA contamination. Regarding frozen samples, 20 mg of tissue were first homogenised in the specific lysis buffer by 6300 r.p.m. 2 × 30 s using Precellys 24 (Bertin technologies), then the RNeasy Mini Kit (Qiagen) was used.

cDNA and quantitative RT-PCR analysis: cDNA was synthesised using a Reverse Transcription Kit (Bio-Rad). PCR reactions were performed using 2X SYBR Green qPCR SuperMix-UDG with Rox (ThermoFisher Scientific) and specific primers (1 µM; qPCR primer sequences are listed in Supplementary Data S11). Expression levels were quantified using the comparative Ct method ($2^{-\Delta\Delta CT}$ method) with the house-keeping gene $Hprt$ as a control for internal normalisation, and results are expressed as $RQ = 2^{-\Delta\Delta CT}$.

**High-throughput sequencing**. Comparative Genomic Hybridisation analysis: Genomic DNA form dissected $Alb$-$R26^{Met}$ tumours ($n = 16$) and control livers ($n = 8$) was analysed by the "Plateforme Biopuces et Sequencage IGBMC" (Illkirch, France) using an Agilent Oligonucleotide Array-Based CGH for Genomic DNA Analysis (CGH microarray 4 × 180 K).

Genome-wide DNA methylation analysis: Methyl-MiniSeq EpiQuest genome-wide sequencing was perform using genomic DNA from dissected $Alb$-$R26^{Met}$ tumours ($n = 10$) and control livers ($n = 3$) to analyse the DNA methylation profile by the Zymo Research Corporation (Irvine, CA, USA).

Library construction. Libraries were prepared from 200–500 ng of genomic DNA digested with 60 units of TaqαI and 30 units of MspI (NEB) sequentially, then extracted with Zymo Research DNA Clean and Concentrator™-5 kit (Cat#: D4003). Fragments were ligated to pre-annealed adapters containing 5′-methyl-cytosine instead of cytosine according to Illumina's specified guidelines (www.illumina.com). Adaptor-ligated fragments of 150–250 bp and 250–350 bp in size were recovered from a 2.5% NuSieve 1:1 agarose gel (Zymoclean™ Gel DNA Recovery Kit, Zymo Research Cat#: D4001). The fragments were then bisulfite-treated using the EZ DNA Methylation-Lightning™ Kit (Zymo Research, Cat#: D5020). Preparative-scale PCR was performed and the resulting products were purified (DNA Clean and Concentrator™–Zymo Research, Cat#D4005) for sequencing on an Illumina HiSeq.

Alignments and data analysis. Sequence reads from bisulfite-treated EpiQuest libraries were identified using standard Illumina base-calling software and then analysed using a Zymo Research proprietary analysis pipeline, which is written in Python and used Bismark (http://www.bioinformatics.babraham.ac.uk/projects/bismark/) to perform the alignment. Index files were constructed using the Bismark-genome-preparation command and the entire reference genome. The non-directional parameter was applied while running Bismark. All other parameters were set to default. Filled-in nucleotides were trimmed off when doing methylation calling. The methylation level of each sampled cytosine was estimated as the number of reads reporting a C, divided by the total number of reads reporting a C or T (β-value).

Overall sequencing results (for 13 samples) are: (a) mean total read: 30 million read pairs, (b) mean mapping efficiency: 40%, (c) mean unique CpGs: 4.1 millions, (d) mean average CpG coverage: 16×, (e) mean bisulfite conversion rate: 98%. Data accessibility: Methylome datasets generated from this study are deposited with the Gene Expression Omnibus (accession GSE90093).

Identification of differentially methylated CpGs. A total of 1.085.757 unique single CpG sites, present in all samples, were analysed. β-value ranged from 0 (not methylated) to 1 (fully methylated). To identify differentially methylated CpGs, the methylation difference per CpG was calculated as the mean β-value of tumours minus the mean β-value of controls. Those with a methylation difference > 0.2 were filtered to retain the ones with a FDR < 0.05 (Student's two-sided $T$-test and Benjamini–Hochberg False Discovery Rate for $P$-value correction). A CpG is classified as "hypomethylated" when the methylation difference is < −0.2 and as "hypermethylated" when the methylation difference is > 0.2. A global analysis was first carried out with all measured CpGs, then dividing the CpGs according to their location within or outside a CGI (CpG Island bedfile downloaded from UCSC). According to the Methyl-MiniSeq EpiQuest coverage, the CGI coverage by CpGs was 87.5%. Studies were focused on CGI regions. The overlap with CGI and the annotated gene was performed using the CGI track from the UCSC genome browser, and Refseq gene annotations based on the NCBI37/mm9 mouse reference. We discarded "ubiquitous CpGs" located in more than one annotated gene, and we extended the gene/CGI annotation to the gene's promoter region to −1.5 kb upstream the TSS.

Targeted Bisulfite Sequencing: Genomic DNA from $Alb$-$R26^{Met}$ tumours dissected from mice treated with Decitabine (2.5 mg/kg; twice per week, for a total of three treatments; $n = 4$) and without treatment ($n = 2$) was used to asses CpG methylation levels in selected regions within the candidate CGIs through bisulfite sequencing by the Zymo Research Corporation (Irvine, CA, USA).

Assay Design, Sample Preparation and Multiplex Targeted Amplification. After assesment of DNA concentration and quality, DNA samples were bisulfite converted using the EZ DNA Methylation-Lightning™ Kit (ref Cat#D5030) according to the manufacturer's instructions. Primers were designed with Rosefinch, Zymo Research's proprietary sodium bisulfite converted DNA-specific primer design tool (primer sequences are listed in Supplementary Data 5). Multiplex amplification of all samples using the specific primer pairs and the Fluidigm Access Array™ System was performed according to the manufacturer's instructions. The resulting amplicons were pooled for harvesting and subsequent barcoding according to the Fluidigm instrument's guidelines. After barcoding, samples were purified (ZR-96 DNA Clean and Concentrator™ –ZR, Cat#D4023), then prepared for parallel sequencing using a MiSeq V2 300 bp Reagent Kit and paired-end sequencing protocol, according to the manufacturer's guidelines.

Targeted Sequence Alignments and Data Analysis. Sequence reads were identified using standard Illumina base-calling software and then analysed using a Zymo Research proprietary analysis pipeline, which is written in Python. Sequence reads were aligned back to the reference genome using Bismark (http://www.bioinformatics.babraham.ac.uk/projects/bismark/), an aligner optimised for bisulfite sequence data and methylation calling[44]. Paired-end alignment was used as default thus requiring both read 1 and read 2 be aligned within a certain distance, otherwise both read 1 and read 2 were discarded. Index files were constructed using the bismark_genome_preparation command and the entire reference genome. The non-directional parameter was applied while running Bismark. All other parameters were set to default. The methylation level of each sampled cytosine was estimated as the number of reads reporting a C, divided by the total number of reads reporting a C or T.

Transcriptome analysis by RNA-seq: Total RNA from dissected $Alb\text{-}R26^{Met}$ tumours ($n = 4$) and control livers ($n = 4$) was processed for transcriptome analysis. RNA quality was controlled using the Agilent RNA 6000 Pico Kit and Agilent 2100 Bioanalyzer (Agilent Technologies, Santa Clara, California) according to the manufacturer's recommendations. Total RNA (1 μg per sample) was used for library preparation using the TruSeq RNA Sample Preparation Kit (Illumina) by GATC Biotech (Mulhouse; NGSelect service). Sequencing was performed on a HiSeq 2500 (Illumina; $2 \times 50$ bp paired end) and base calling performed using RTA (Illumina). Quality control of raw reads was done using FastQC (http://www.bioinformatics.babraham.ac.uk/projects/fastqc/). Reads were mapped to the reference genome mm9 with STAR aligner[45] using default parameters; differential expression was calculated using the Cufflinks package[46].

**Cell culture-related experiments**. Cell lines: $Alb\text{-}R26^{Met}$ HCC cell lines (HCC3, HCC13 and HCC14) were established, characterised and cultured as previously described[24]; cells were regularly tested by PCR-based assay to confirm their maintenance in free *Mycoplasma* culture condition.

shRNA-mediated downregulation of candidate genes: The functional relevance of candidate oncogenes was determined using shRNA targeting sequences (Sigma; shRNA sequences are reported in Supplementary Data 12). In particular, plasmids carrying the shRNA sequence were transfected in cells using Lipofectamine 2000 reagent, according to the manufacturer' instructions (ThermoFisher Scientific). After 1 week of puromycin selection, pools of resistant clones were used to verify downregulation of gene expression levels (by RT-qPCR and western blots) and to perform biological assays.

Cell drug treatment: For the demethylation experiments, cells were exposed to 0.3 μM of Decitabine (5-Aza-2'-deoxycytidine; Selleckchem) for 48 h. After treatment, cells were used for cell viability, anchorage-dependent growth assay, anchorage-independent growth assay, tumour sphere formation assay and xenograft studies. For experiments shown in Supplementary Fig. 14, after 48 h of Decitabine treatment, cells were cultured for 10 days with complete media before performing the experiments.

Survival assay: Cells were seeded in a 150-μl volume per well in 96 well plates (10,000 cells/well) in 10% serum for 24 h, then Decitabine treatment was applied at 0.3 μM in the corresponding wells. After 48 h, cell viability was assessed in a Cell Titer Glo Luminescence Assay (Promega) and luminescent signals were measured with a luminometer microplate reader (Berthold). Data are expressed as means ± SEM of three independent experiments performed in triplicate.

Anchorage-dependent growth assay (focus formation assay): To measure anchorage-dependent growth, 300 cells were seeded in 10 ml complete media in a 10 cm dish. After 7 days, foci were stained with a 0.2% crystal violet solution (2% methanol). The total number of foci and individual foci size were quantified using ImageJ program. Data are expressed as means ± SEM of three independent experiments performed in triplicate.

Anchorage-independent growth assay (soft agar assay): Assays were performed as previously described[47–49]. Briefly, cells were cultured in 12-well plates containing two layers of agar. Cells ($6 \times 10^3$) were resuspended in 0.5% agar diluted in complete medium and poured onto a 1% layer of agar (diluted in medium). Fresh medium was added to the top layer every 3 days. After 2 weeks, colonies were stained with MTT, pictures were taken using a dissecting microscope, and colonies were counted using ImageJ software. Numbers are expressed as means ± SEM of three independent experiments performed in triplicate.

Tumour sphere forming assay: Cells were cultured at a density of $2 \times 10^4$/ 35 mm dishes in a stem cell-permissive media. In particular, cells were cultured for one week in DMEM/F12 medium supplemented with 1% N-2 Supplement, 2% B27, 50 mg/ml of Penicillin-Streptomycin, glutamine (Gibco), 0.01% Bovine Serum Albumin (BSA), 5 mg/ml of insulin (Sigma) and growth factors including 10 ng/ml of basic fibroblast growth factor (bFGF), 20 ng/ml of epidermal growth factor (EGF) and 10 ng/ml of hepatocyte growth factor (HGF; Peprotech). After one week, pictures of the whole dish were taken using a dissecting microscope, and spheres were counted using ImageJ software. Numbers are expressed as means ± SEM of three independent experiments performed in triplicate.

In vivo tumorigenesis assays (xenografts in nude mice): For in vivo demethylation studies, xenografts were performed using $Alb\text{-}R26^{Met}$ HCC cells either untreated or pre-treated for 48 h with Decitabine (0.3 μM). Cells ($5 \times 10^6$) were then resuspended in a 1:1 Matrigel:PBS solution (Corning BV) and inoculated subcutaneously into the flank-leg region of nude mice (S/SOPF SWISS NU/NU; Charles River). After 5 days of cell inoculation, mice were treated with

intraperitoneal injections of vehicle or Decitabine (2.5 mg/kg) twice per week for 3 weeks. Mice were then sacrificed and tumour volume was measured as length × width × height. For assessment of in vivo tumorigenic capacity of candidate genes, xenografts were performed using $Alb\text{-}R26^{Met}$ HCC cells ($1 \times 10^6$) either un-transfected, transfected with shControl, or with a shRNA sequence targeting the candidate gene. Tumour volume was followed every week. After 6 weeks mice were sacrificed and tumour volume after dissection was determined as length × width × height.

Western blots: Protein extracts from HCC cells were prepared and western blot analysis was performed as previously described[43,48,49]. For SCN8A detection, protein lysates were run on a 5% SDS gel and transferred overnight at 300 mA in the presence of 0.1% SDS. The acquisition of ECL signal was performed using the MyECL imager system (ThermoFisher Scientific)(Supplementary Fig. 15).

Antibodies: Antibodies used were: anti-SCN8A (Abcam, #ab65166; 1:500), anti ACTN-1 (Cell Signalling, #6487; 1:3000), anti-SRD5A2 (ThermoFisher Scientific, #PA5-25465; 1:1000), anti-NFkB2 (Cell Signalling; #4882; 1:1500), anti-NEURL1B (Abcam, #ab156988; 1:3000), anti-ACTIN (Sigma, A3853; 1:5000), anti-rabbit IgG-peroxidase or anti-mouse IgG-peroxidase (Jackson; 1:4000).

**Computational analyses**. Unsupervised hierarchical clustering analysis: Clustering statistics was determined by using the methylation values of all CGIs for each sample. We applied the Principal Component Analysis and the Agglomerative Distance Tree using the "linkage" function with unweighted average euclidean distance for calculating the similarity matrix of samples and the "dendrogram", as well as "phylotree" function to plot the hierarchical and distant trees (both are from Matlab Statistical Toolbox). For studies reported in Supplementary Fig. 7, clustering analysis of both methylome and expression data was performed using the function "hclust" on an Euclidean distance matrix of samples, which was computed with the function "dist". "hclust" then returned a tree-like structured object that could be plotted as dendrogram by "plot" (R, version 3.3.1).

Identification of human CGIs corresponding to the mouse CGIs of interest: To compare methylome outcomes identified in the $Alb\text{-}R26^{Met}$ genetic system with those available for human studies, genomic coordinates were converted from mm9 to GRCh37/hg19 by using the "Lift-Over" tool available from UCSC (https://genome-euro.ucsc.edu/cgi-bin/hgLiftOver). This allowed us to successfully map 501 out of 513 CGIs from mouse to human regions (Supplementary Data 2). Among them, we only kept 501 unique human regions by discarding duplicate lift-overs. We also discarded 14 human regions not overlapping with any human CGI. We then check into TCGA patient datasets the presence of methylation data for those CGIs. We focused the analysis on the patient having both tumour and control samples (adjacent liver) methylation data, and we discarded the CGIs having no entry into any of the TCGA patient dataset. Finally, the total CGIs used for comparative analyses between mouse and human is 416.

Analysis of public available DNA methylome data: The human methylome data is available through firebrowse (www.firebrowse.org) by the BROAD Institute and is based upon data generated by the TCGA Research Network: http://cancergenome.nih.gov/. The publicly available methylome data (Level 3 data) of HCC patients from TCGA is generated with the platform Illumina Infinium Human DNA Methylation 450 and contains beta values for 485778 CpGs. Patients with both tumour and control samples were extracted and calculation of methylation difference per CpG was applied (β-values of tumour–β-values of control). Student's $T$-test was used to compare between tumour and normal samples, and the $P$-values were corrected with Benjamini–Hochberg False Discovery Rate (FDR). As our methylome screen focused on CGIs, we revisited the human data (from TCGA and from GSE56588) to generate a list of all CpGs within CGIs with the corresponding methylation β-values. By applying the same methylation difference and FDR thresholds used for $Alb\text{-}R26^{Met}$ methylome data, we extracted a list of differentially methylated CpGs from the human HCC dataset.

Methylome overlap between $Alb\text{-}R26^{Met}$ outcomes and human data: A methylation overlap between $Alb\text{-}R26^{Met}$ and human HCC was considered only when a given CGI was differentially methylated in both species. To define the methylation status of a given CGI, the CpG with an absolute maximum methylation difference among all patient samples was chosen as a representative probe (with $P$-value threshold and fold change cut-off defined above). This CpG was analysed in all HCC patients. An overlap score (in percentage) was determined by calculating the number of human CGIs differentially methylated versus the total number of lifted-over CGIs subset.

Analysis of public available RNA-seq data: The human RNA-seq data from TCGA was available through firebrowse. The data is generated with the platform Illumina HiSeq 2000 Sequencing System and uses MapSplice[50] to do the alignment and RSEM[51] to perform the quantitation. The scaled estimate from RSEM output was used as this value could be multiplied by $10^6$ to obtain a measure in terms of transcripts per million (TPM), which is preferred over RPKM[52] or FPKM[53] as it is independent of the mean transcript length and therefore more comparable across samples[51]. The TPM is calculated for each gene and the calculation of $Log_2$ Fold Change ($Log_2$(tumour sample)−$Log_2$(control sample)) was applied to each patient with available data from both tumour and control samples.

Calculation of the relative position to the ATG: For calculating the position of CpGs, we used the longest transcript for each gene. The gene length was reported with values ranging from −100% and +100% (transcription end site: TES), with the ATG at position 0. The relative position for each CpG was then reported

relative to its distance to the ATG. A positive relative position corresponds to a genomic region located downstream the ATG, whereas a negative relative position stands for a genomic region located upstream the ATG.

Analysis of public available data from a mouse HCC model carrying the viral hepatitis B virus X expression: Using available methylome and expression data based on a HCC model induced by the viral hepatitis B virus X (*HBx^{tg}*; GSE48052[27]), we performed the same analysis done for the *Alb-R26^{Met}* model (Fig. 3b). For each CpG, the methylation difference between *HBx^{tg}* tumour and control sample was calculated as the difference of the RPKM. For those CpGs found differentially methylated, the expression of the corresponding gene was then calculated as the difference of the RPKM sum within the TSS and TSE.

Pathway enrichment analysis: For these analyses (shown in Supplementary Figs. 5, 9), identified genes were used as an input for KEGG pathway enrichment analysis with the REST API tool (http://rest.kegg.jp). Pathways were further ranked by $-\log_{10}$ P-value after applying the hypergeometric probability density function (Matlab function "hygepdf" from Statistical Toolbox).

Statistical analysis: All data were analysed using GraphPad Prism software (version 7.01) and Matlab Statistical Toolbox (version R2015b). Results are expressed as the median (indicated by a line) or as the mean ± standard error of the mean (SEM), according to sample distributions. Statistically significant differences were estimated by applying unpaired Student *t*-tests to data showing normal distributions, and Mann–Whitney tests in all other situations. Moreover, one-way-ANOVA was used to determine differences between the means of independent groups (in vivo xenograft experiments in Figs. 2h and 7d), and Fisher's exact test for categorical variables (risk factors in Supplementary Fig. 9). All statistical tests were two-sided. Statistical significance was defined as not significant (ns): $P > 0.05$; *$P < 0.05$; **$P < 0.01$; ***$P < 0.001$. Significance is indicated in figures

**Data availability**. Raw and processed data of bisulfite sequencing have been deposited to the Gene Expression Omnibus (GEO) [GEO: GSE90093]. The authors declare that all data supporting the findings of this study are available within the article and its Supplementary Information files, or from the authors upon reasonable request.

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

## Acknowledgements

These results are in part based upon public data generated by TCGA Research Network: http://cancergenome.nih.gov/. We are particularly grateful to F. Helmbacher for extremely valuable feedback on the study. We thank: all members of our labs for helpful discussions and comments; F. Castets for advises on biochemical analysis; J. Zucman-Rossi and E. Letouzé for valuable discussion and feedback on the bioinformatics; V. Girod-David and L. Jullien for excellent help with mouse husbandry; C. Giaccherini, A. Dobric and E. Chrabaszcz for their contributions to studies on molecular and functional characterisations of shRNA-targeted HCC cells. This work was funded by INCa (Institut National du Cancer), FdF (Fondation de France) and GEFLUC–Les Entreprises contre le Cancer to F.M. M.A. was supported by a FdF fellowship. A.Y. was supported by DFG grant HA 6905/2-1. S.K.B. was supported by the Higher Education Commission (HEC) of Pakistan–France Campus. The contribution of the Region Provence Alpes Côtes d'Azur and of the Aix-Marseille Université to the IBDM animal facility is also acknowledged. The funders had no role in study design, data collection and analysis, decision to publish or preparation of the manuscript.

## Author contributions

M.A.: performed the majority of the experiments, data analysis, interpretation, contributed to computational work and writing the paper. F.D.: performed the majority of the computational work, data analysis and interpretation. A.Y.: performed computational work using the human HCC cohort from TCGA database, data analysis and interpretation. S.K.B.: contributed to molecular and functional studies with HCC cells. S.R.: prepared tumour samples and genomic DNA for methylome; contributed to xenograft studies. R.D.: contributed to establishing the mouse model, to interpreting data and provided input on writing the paper. A.J.S.: provided support for computational work, contributed to interpreting data and provided input on writing the paper. B.H.H.: analysed and interpreted transcriptome data from mouse model, supervised studies on human HCC data and provided input on writing the paper. F.M.: designed the study, performed experiments, analysed and interpreted data, ensured financial support and wrote the paper.

## Additional information

**Competing interests:** The authors declare no competing interests.

