## [Peer Review File · Nature Communications]

Reviewers' comments:

Reviewer #1 (Remarks to the Author):

The paper by Arechederra et al. identifies a subset of genes that are hypermethylated and overexpressed in a murine model of hepatocellular carcinoma (HCC), which corresponds to potential oncogenes driving the disease. By combining a sophisticated mouse model of HCC, DNA methylation and expression profilings, and information from human samples, the authors identify changes in methylation/expression that are common in both mouse and human HCCs. The findings reported in this manuscript are potentially interesting. However, there are several concerns that need to be addressed to consider publication in Nature Communications:

Main concerns:

- In Figure 1, 513 CpG islands (CGIs) differentially methylated in murine HCCs are found and some of those CGIs are also hypermethylated in an HCC subgroup that is defined as HCC subgroup 3 in two different human datasets. Is Met, the driving oncogene in the murine model, overexpressed or amplified in this HCC subgroup 3 compared to the other 2 subgroups? Is this HCC subgroup 3 characterized by some Met-related gene signature? One would expect that this is the case. If it's not, it would be important to discuss it in the paper.

- In Figure 2, a DNA demethylating agent, decitabine, is used to show that murine HCC cell lines established from the original model are sensitive and display reduced in vitro and in vivo proliferation upon treatment. It would be important to check the effect of this drug in "normal" human or murine hepatocyte lines to demonstrate that changes in DNA methylation are behind this dependency.

- In Figure 4, two different patient subsets are defined: H+E+ and no H+E+. Which is the overlapping of the genes that are H+E+ in humans and in mice? Is Met overexpressed or amplified in this patient population? Is there any overlap between HCC subgroup 3 (Fig. 1) and H+E+ patients? All this should be tested and discussed in the paper.

- In Figure 5, the functional relevance of several genes is tested by performing knockdown experiments in the murine HCC cells. Knocking down the candidate genes affects the proliferation of the cancer cells, indicating that they are required for the proliferation or survival of the cancer cells. However, one cannot conclude that these genes are actually oncogenes since that would involve performing experiments addressing tumorigenesis rather than effects of cell proliferation. The text throughout the manuscript should be modified to reflect that these genes affect tumor cell proliferation rather than being oncogenes. For example, the sentence: "Among reprogrammed genes, whereas several are well-known oncogenes, for others not previously linked to cancer we demonstrate here their action as oncogenes" should be modified, among other sentences throughout the paper. Moreover, experiments using a control shRNA and at least 2 independent shRNAs (rather than only one) should be performed to rule out off target effects. Finally, the expression of a subset of these candidate genes should be tested in the murine HCC cells treated with decitabine, to see whether their expression is affected (downregulated) by DNA demethylation. If that's the case, changes in DNA demethylation should be verified by bisulfite sequencing of the candidate CGIs.

Minor concerns/suggestions:

- "Alb-R26Met mice model" change to "Alb-R26 mouse model". In Fig. 2E, use "mouse weight" instead of "mice weight".

- The meaning of the word "unicity" may not be obvious for most readers.

- Cdkn2a is a well-known tumor suppressor gene that you have found methylated and overexpressed (Fig. 3F). It would be important to check whether the same phenomenon is observed in some human samples since CDKN2A promoter hypermethylation is a frequent mechanism of gene silencing in HCC. It would also be important to discuss in the manuscript.

Reviewer #2 (Remarks to the Author):

In the manuscript "Hypermethylation of gene body CpG islands predicts high dosage of functional oncogenes in liver cancer", Arechederra et al., investigated impact of DNA methylation on transcriptional switches associated with hepatocarcinogenesis. Using HCC mouse model (Alb-R26Met) and publically available data sets collected from human HCC samples, they observed enrichment of genes with simultaneous hypermethylation of CpG islands within 5' UTR or in gene body region, and upregulation of expression. Interestingly, their overexpression positively correlates with the CGI distance to the ATG. Furthermore, they observed that patients, who belong to H+ E+ subset, belong to aggressive "proliferative progenitor" HCC patient subset. The authors propose existence of the "oncogenic module" where set of over-expressed oncogenes work together to promote tumorigenesis.

This is very interesting study, potentially with the high impact in the field, however, some conclusions are not sufficiently supported by the current data and additional experiments and changes in the manuscript are needed before manuscript is suitable for publication.

Specific comments:

1. To analyze methylome and expression profile of the HCC in Alb-R26Met mouse model, the authors use wt liver as a control. Generally, as a control, non-tumor regions of the same livers (same genotype) are used. Is there a reason why the authors chose to use wt liver as a source of control DNA and RNA?
2. As a source of DNA and RNA for their analyses, the authors use crude tumor/liver extract, mixture of many different cell populations, including hepatocytes, chondrocytes, stellate cells, immune cells, endothelial cells (liver) or tumor cells, immune cells, TAF (tumor). That heterogeneity can significantly alter results. Did the authors consider using pure (sorted) cell populations for their studies?
3. The authors observed very nice correlation between data obtained in Alb-R26Met mouse model and analyzed patient samples. What about other HCC mouse models?
4. In the Figure 2, the authors show that Decitabine treatment decreases anchorage independent growth in three analyzed cell lines and they conclude that demethylation decreases tumorigenic potential of the cells. This is too strong conclusion based only on one assay, soft agar. How about colony formation assay, proliferation, viability, senescence, cancer stem cells presence? Are they affected after Decitabine treatment?
5. To investigate Decitabine effect in vivo, the authors use xenograft mouse model. Being that nude mice have impaired immune system (that can alter drug response), it would be more clinically relevant if for the treatment orthotopic model was used or Alb-R26Met mice with developed liver tumors. Because mouse HCC cell lines have been used, they should be able to grow in wt mice when implanted in the liver.
6. After performing anchorage independent assay with the cells that have silenced Scn8a, AdamtsL5 or Actn1, the authors conclude that silencing of over-expressed genes decreases tumorigenic properties of the cells. This is strong statement based on only one assay. Additional experiments with silenced cells should be performed to solidify that conclusion, including investigating in vivo tumorigenic potential using orthotopic mouse model.
7. What causes changes in global DNA methylation in tumor cells? Is DNMT3B affected in Alb-R26Met hepatocytes?
8. In the line 232, the authors state that "overall survival of H+E+ versus "NO H+E+" patient subsets showed a trend of better prognosis for H+E+ patients". The same patients belong to

aggressive “proliferative-progenitor subclass”, that have poorer prognosis. Can authors comment on that?

Minor points:

1. In the Figure 2B and C, if the cells are left in the culture for ten days or longer after the last pulse with Decitabine, do they recover?
2. Figure 5E, silencing efficiency should be demonstrated on the protein level, not only mRNA level as shown in the Supplemental File.
3. Figure 5E, why are Scn8a, AdamtsL5 and Actn1 chosen to be investigated?

Rebuttal letter to the referees.

We thank the two reviewers for their valuable suggestions and extremely constructive criticisms on our studies. We undertook a thorough revision of the manuscript. Altogether, the recommended additional analysis and experimental studies reinforce the relevance of an epigenetic reprogramming of an “oncogene module” through hypermethylation in gene body CGIs.

Reviewer #1:

The paper by Arechederra et al. identifies a subset of genes that are hypermethylated and overexpressed in a murine model of hepatocellular carcinoma (HCC), which corresponds to potential oncogenes driving the disease. By combining a sophisticated mouse model of HCC, DNA methylation and expression profilings, and information from human samples, the authors identify changes in methylation/expression that are common in both mouse and human HCCs. The findings reported in this manuscript are potentially interesting. However, there are several concerns that need to be addressed to consider publication in Nature Communications:

Main concerns:

- In Figure 1, 513 CpG islands (CGIs) differentially methylated in murine HCCs are found and some of those CGIs are also hypermethylated in an HCC subgroup that is defined as HCC subgroup 3 in two different human datasets. Is Met, the driving oncogene in the murine model, overexpressed or amplified in this HCC subgroup 3 compared to the other 2 subgroups? Is this HCC subgroup 3 characterized by some Met-related gene signature? One would expect that this is the case. If it's not, it would be important to discuss it in the paper.

Authors' response: We agree with the reviewer about the relevance of analysing *Met* levels in the three human HCC subgroups identified by the *Alb-R26^{Met}* methylome screening. Concerning the HCC patient cohort from GSE56588, expression data (array) are only publicly available for some patients and without information about mutations. Therefore, correlative studies are not possible with this HCC cohort. Instead, we were able to perform correlative studies on the HCC cohort from TCGA as RNA-seq and mutation data are available. In particular, we analysed *Met* mutations and *Met* expression levels for each patient belonging to the 3 different HCC subgroups (Figure 1F). All HCC patients carry the wild-type form of *Met*, which is in agreement with rare mutations of *Met* in HCC. Concerning expression levels, *Met* is overexpressed in 86% (6/7) of HCC patients belonging to subgroup 3 (the subgroup that best overlap with CGI methylation changes in *Alb-R26^{Met}*), 32% (6/19) to HCC subgroup 2, and only 13% (2/15) to HCC subgroup 1. Patients with *Met* overexpression are highlighted with a red dot in Figure 1F and additional information about the correlation between *Met* levels and percentage of overlap with *Alb-R26^{Met}* CGI methylation changes are reported in Supplementary Figure S3C-F of the revised version of the manuscript. Collectively, these additional analyses highlight a striking correlation between differentially methylated CGIs and *Met* overexpression in the HCC patient subgroup 3, which is well modelled by the *Alb-R26^{Met}* genetic setting.

- In Figure 2, a DNA demethylating agent, decitabine, is used to show that murine HCC cell lines established from the original model are sensitive and display reduced in vitro and in vivo proliferation upon treatment. It would be important to check the effect of this drug in "normal" human or murine hepatocyte lines to demonstrate that changes in DNA methylation are behind this dependency.

Authors' response: We addressed this reviewer's issue with a set of experiments (also related to point 4 of reviewer 2). First, we just want to precise to the reviewer that the experiments showed in Figure 2 in the previous version assessed cell tumorigenicity in vitro (anchorage independent growth; also known as soft agar assay) and in vivo (xenograft in nude mice) and not cell proliferation. Anchorage-independent growth is the ability of transformed cells to grow independently of a solid

surface, and is a hallmark of carcinogenesis. This is a well-established method to ascertain tumorigenicity *in vitro* and is considered one of the most stringent tests for malignant transformation in cells (for example, in: *Stanley Borowicz, 2014, J Vis Exp*). As “normal” human or murine primary hepatocytes do not proliferate, we addressed the question raised by the reviewer by exploring the effects of Decitabine on cell viability of *Alb-R26^{Met}* HCC cells as well as of mouse liver progenitor MLP-29 cells, which are not tumorigenic as shown by their incompetence to form colonies in soft agar (*Giordano et al. Nature Cell Biology 2002*). We found that Decitabine treatment does not affect cell viability of cancer cells (*Alb-R26^{Met}* HCC) and of “normal” cells (MLP-29). Thus, changes in DNA methylation triggered by Decitabine impact the tumorigenic properties of cancer cells (as previously shown by anchorage-independent growth assays and xenografts in nude mice, and in the revised version as well by anchorage-dependent growth assays, sphere formation assay, xenografts in nude mice) rather than impacting cell viability. All these data have been included in the revised version of the manuscript (Figure 2).

- In Figure 4, two different patient subsets are defined: H+E+ and no H+E+. Which is the overlapping of the genes that are H+E+ in humans and in mice? Is Met overexpressed or amplified in this patient population? Is there any overlap between HCC subgroup 3 (Fig. 1) and H+E+ patients? All this should be tested and discussed in the paper.

Authors’ response: Concerning the overlapping of genes between human and *Alb-R26^{Met}* mouse model, in our previous version of the manuscript we reported for the 41 TCGA HCC patients, the expression level of 42/55 genes found hypermethylated and overexpressed in the *Alb-R26^{Met}* tumours. Re-analysing the transcriptome of each patient, expression data were available for 51/55 genes (no expression data are available for 4 genes: *Mt1*, *Prickle2*, *Neurl1a*, and *Tmem191c*; Figure 6A-C in the revised version). We further extended our previous studies by analysing as well in each HCC patient the methylation levels of the CGIs corresponding to the 55 genes. We took into account that the number of CGIs for each gene varies between genes (reported in Supplementary Table S7 and S8). 53/55 genes successfully lifted-over between mouse and human (not lift-over for *Mt1* and *Prickle2*) and both methylation and expression data are available for 51 genes (no methylation data are available for *Tmem191c* and *Neurl1a*; reported in Supplementary Table S8). These additional analyses revealed that 42/51 (82%) genes are both hypermethylated and overexpressed in at least 1 patient, and that 40/41 (97,5%) patients have at least 1 genes both hypermethylated and overexpressed (Figure 6D, E in the revised version). Furthermore, there is a significant higher number of genes both hypermethylated and overexpressed in the H+E+ patient subset compared to the “No H+E+” subset (“H+E+” vs “No H+E+”: p value < 0.001). All these findings are reported in Figure 6D, E, and Supplementary Figure S11 and Supplementary table S8 in the revised version of the manuscript.

Concerning the issue of whether Met is overexpressed or amplified in the H+E+ patient subset compared to the “NO H+E+” subset, we performed a series of analyses. We found that the mean Met levels in the H+E+ subgroup is $0,77 \pm 0,16$ (9/23 (39%) patients with Met levels ≥ 1), whereas in the “No H+E+” subgroup is $0,2 \pm 0,24$ (5/18 (27%) patients with Met levels ≥ 1). Although there is a trend in correlating Met increased levels with the H+E+ patient subset, this correlation is not significant. One possible explanation could be that CGI hypermethylation and gene overexpression in the H+E+ patient subset is ensured by a set of signals rather than merely by Met. As additional thoroughly studies would be required to address this issue, we favour not to include these data in the revised version of the manuscript. Nevertheless, we provide them as Figure 1 for the Reviewer.

Concerning a putative overlap between HCC subgroup 3 (in Figure 1) and H+E+ patients, 5/7 patients of HCC subgroup 3 belong to the H+E+ subset. Interestingly, all of these 7 patients are characterized by more than 37% of genes both hypermethylated and overexpressed (highlighted with a red square and red % in Figure 6A in the revised version of the manuscript). Detail information is reported as well in Supplementary Table S56.

- In Figure 5, the functional relevance of several genes is tested by performing knockdown experiments in the murine HCC cells. Knocking down the candidate genes affects the proliferation of the cancer cells, indicating that they are required for the proliferation or survival of the cancer cells. However, one cannot conclude that these genes are actually oncogenes since that would involve performing experiments addressing tumorigenesis rather than effects of cell proliferation. The text throughout the manuscript should be modified to reflect that these genes affect tumor cell proliferation rather than being oncogenes. For example, the sentence: "Among reprogrammed genes, whereas several are well-known oncogenes, for others not previously linked to cancer we demonstrate here their action as oncogenes" should be modified, among other sentences throughout the paper. Moreover, experiments using a control shRNA and at least 2 independent shRNAs (rather than only one) should be performed to rule out off target effects.

Authors' response: We agree with the reviewer that additional studies were needed to strengthen whether hypermethylated and overexpressed genes act within an "oncogenic module" to ensure tumorigenicity. We consequently performed a new set of experiments. As above, we just precise to the reviewer that the experiments reported (Figure 5E in the previous version; Figure 7A in the revised version) assess in vitro tumorigenicity (anchorage independent growth; also known as soft agar assay) rather than cell proliferation. We further explored the capability of selected genes to attribute cell tumorigenicity by performing as well: 1) anchorage dependent growth assays (also known as foci formation assay; 2) tumour sphere formation assays; 3) xenografts in nude mice. All results from these additional studies are included in the revised version of the manuscript (Figure 7B-D). Collectively, these studies demonstrate that downregulation of each candidate gene in *Alb-R26^{Met}* HCC cells interfere with their tumorigenic properties.

We have selected 3 new candidates belonging to the list of genes both hypermethylated and overexpressed in *Alb-R26^{Met}* tumours (*Srd5a2*, *NFkB2*, and *Neurl1b*). In addition to the 2 genes reported in the previous version (*Scn8a* and *Actn1*), for all the 5 genes we have generated mHCC cells stably transfected with shRNA targeting sequence (using 2 different shRNAs) and performed all function studies mentioned above: anchorage independent assays, anchorage dependent assays, tumour sphere formation assays, and xenografts in nude mice. Collectively these studies show that each of these five genes contribute to cell tumorigenicity, thus acting within an "oncogene module".

Following reviewer suggestion, we included studies using a control shRNA as well as a second shRNA targeting sequence for each candidate gene (Figure 7A and Supplementary Figure S13). In particular, a control shRNA and a second shRNA for each candidate have been transfected to *Alb-R26^{Met}* HCC cells (mHCC14). After selection with puromycin, stable shControl/shCandidate cells were used to perform soft agar assay. Concerning control shRNA, similar colony numbers were observed between mHCC14 and shControl mHCC14 cells. As suggested by the reviewer, the shControl mHCC14 cells have been included as control for all experiments. Moreover, the reduction of colony numbers by targeting a given candidate gene is supported by results using two different shRNA targeting sequence. All these data have been included in the revised version of the manuscript (Figure 7 and Supplementary Figure S13 of the revised version of the manuscript).

Finally, the expression of a subset of these candidate genes should be tested in the murine HCC cells treated with decitabine, to see whether their expression is affected (downregulated) by DNA

demethylation. If that's the case, changes in DNA demethylation should be verified by bisulfite sequencing of the candidate CGIs.

Authors' response: Following reviewer suggestion, we analysed whether Decitabine treatment would affect the expression levels of a set of genes. We decided to address this reviewer's point directly in vivo as we reasoned that outcomes would be more relevant, although the experimental setting rather challenging. Therefore, for a set of genes found hypermethylated and overexpressed in the *Alb-R26^{Met}* tumours, we analysed both their expression levels and the methylation levels of their corresponding CGIs in dissected tumours from *Alb-R26^{Met}* mice either untreated or treated with Decitabine. RT-qPCR results show that Decitabine treatment significantly decreased the expression levels of 7 out of 8 analysed genes. Bisulfite sequencing analyses revealed a decreased methylation levels of most CpGs within the corresponding CGIs. Concerning the *Scn8a* gene, the methylation levels of its gene body CGI was reduced in Decitabine treated tumours compared to untreated tumours. This was accompanied by a trend in downregulation of its expression levels, although not significant. It is possible that for *Scn8a*, the demethylation extent caused by the dose of Decitabine we used is suboptimal to influence its expression levels. Alternatively, a more complex mechanism could be involved in the regulation of its expression. All these data have been included in the revised version of the manuscript (Figure 4).

Minor concerns/suggestions:

- "*Alb-R26^{Met} mice model*" change to "*Alb-R26 mouse model*". In Fig. 2E, use "*mouse weight*" instead of "*mice weight*".

Authors' response: We thank the reviewer for the correction. We have modified the text accordingly.

- *The meaning of the word "unicity" may not be obvious for most readers.*

Authors' response: We have replaced the word "unicity" with "uniqueness".

- *Cdkn2a is a well-known tumor suppressor gene that you have found methylated and overexpressed (Fig. 3F). It would be important to check whether the same phenomenon is observed in some human samples since CDKN2A promoter hypermethylation is a frequent mechanism of gene silencing in HCC. It would also be important to discuss in the manuscript.*

Authors' response: Indeed, *Cdkn2a* is a well-known tumour suppressor; it has been reported that its promoter methylation is a mechanism by which it is downregulated in HCC (Csepregi et al. BMC Cancer 2010). Nevertheless, we intriguingly found in the *Alb-R26^{Met}* cancer model that *Cdkn2a* is overexpressed and hypermethylated in its gene body CGI, whereas no methylation changes were observed in its promoter CGI. To address whether this phenomenon would also occur in HCC patients, we analysed *Cdkn2a* methylation and expression in HCC patients from the TCGA cohort as well as from the GSE56588 cohort (for which methylation and expression data are available: 205/224). Mouse *Cdkn2a* has two CGIs: one in the promoter and another in the gene body. Instead, human *Cdkn2a* has 5 CGIs: one in the promoter and four in the gene body. Data are available only for the CGI in the promoter and for one of the four CGIs located in gene body. Notably, in both cohorts we found an enrichment of patients with an overexpression of *Cdkn2a* (39/41 and 166/204, in the respective cohorts), which is associated to a hypermethylation of the gene body CGI (21/39 and 163/166, in the respective cohorts). In contrast, no methylation changes have been detected in the promoter CGI for both HCC cohorts. Therefore, these analyses confirm an intriguing phenomenon of *Cdkn2a* overexpression and hypermethylation in gene body CGIs both in the *Alb-R26^{Met}* mouse model

as well as in human HCC patients. These data have been included in the revised version of the manuscript (Supplementary Figure S11).

Reviewer #2:

In the manuscript “Hypermethylation of gene body CpG islands predicts high dosage of functional oncogenes in liver cancer”, Arechederra et al., investigated impact of DNA methylation on transcriptional switches associated with hepatocarcinogenesis. Using HCC mouse model (Alb- R26Met) and publically available data sets collected from human HCC samples, they observed enrichment of genes with simultaneous hypermethylation of CpG islands within 5’ UTR or in gene body region, and upregulation of expression. Interestingly, their overexpression positively correlates with the CGI distance to the ATG. Furthermore, they observed that patients, who belong to H+ E+ subset, belong to aggressive “proliferative progenitor” HCC patient subset. The authors propose existence of the “oncogenic module” where set of over-expressed oncogenes work together to promote tumorigenesis.

This is very interesting study, potentially with the high impact in the field, however, some conclusions are not sufficiently supported by the current data and additional experiments and changes in the manuscript are needed before manuscript is suitable for publication.

Specific comments:

1. To analyze methylome and expression profile of the HCC in Alb- R26Met mouse model, the authors use wt liver as a control. Generally, as a control, non-tumor regions of the same livers (same genotype) are used. Is there a reason why the authors chose to use wt liver as a source of control DNA and RNA?

Authors’ response: We have extensively discussed this issue in the lab before performing these screen studies. Our final decision to use wild-type livers instead of adjacent non-tumour region of the same mutant liver was favoured as we could not predict whether and to which extent increased Met expression levels would attribute changes in methylation and expression of set of genes in the liver before tumorigenesis would occur. As the main goal of our study was to identify alterations associated with tumorigenicity, we favoured the use wild-type livers. Nevertheless, the issue raised by the reviewer is an intriguing topic, which we are currently addressing by analysing RNA-seq data from wild-type and *Alb-R26^{Met}* healthy livers. We do not have yet the answer, as we are currently bioinformatically processing RNA-seq data (which will be the topic of a future manuscript). Nevertheless, we performed RT-qPCR analyses of *alpha-fetoprotein (AFP)* and *Glypican3 (GPC3)* (two HCC markers) as well as *Ki67* (a proliferation marker), in wild-type, *Alb-R26^{Met}* healthy livers, *Alb-R26^{Met}* livers adjacent to tumours, and *Alb-R26^{Met}* tumours (as controls). We observed comparable levels for these three genes in wild-type and *Alb-R26^{Met}* healthy livers. Instead, expression levels of *Ki67*, but not of *AFP* or *GPC3*, are significantly increased in *Alb-R26^{Met}* livers adjacent to tumours. These data have been included in the Figure 2 for Reviewer. We think that these results would support the possible appropriateness of using healthy livers rather than livers from adjacent tumours to specifically search for alterations in tumoral versus healthy tissues.

2. As a source of DNA and RNA for their analyses, the authors use crude tumor/liver extract, mixture of many different cell populations, including hepatocytes, chondrocytes, stellate cells, immune cells, endothelial cells (liver) or tumor cells, immune cells, TAF (tumor). That heterogeneity can significantly alter results. Did the authors consider using pure (sorted) cell populations for their studies?

Authors’ response: This is another very interesting issue and we believe that it would be highly relevant to perform similar studies on sorted cells in order to determine changes characterising different cell populations within the tumour mass (e.g. cancer cells, hepatocytes, immune cells, cancer-associated fibroblasts). However, we favoured the use of tumour/liver extracts as most

analyses of human HCC biopsies are performed similarly. We think that a cell sorting approach on mouse samples would possibly limit some correlative studies with available human data, and to certain extent the translation of findings in clinics. Based on outcomes reported in this manuscript, and having identified HCC patients modelled by the *Alb-R26^{Met}* genetic setting, it would be interesting to perform in the future studies suggested by the reviewer using different cell populations after sorting, ideally using mouse and human tumours.

3. The authors observed very nice correlation between data obtained in Alb- R26Met mouse model and analyzed patient samples. What about other HCC mouse models?

Authors' response: This is an interesting topic, although not easy to address as it requires a thoroughly analysis of expression and methylation profiles, similar to those we obtained in our studies and reported in the present manuscript. Following the reviewer suggestion, we searched for available methylation and expression data from other HCC mouse models and found one based on a HCC model induced by the viral hepatitis B virus X (HBV; GSE48052; Lee et al. PNAS 2014). We therefore used these available data and performed the same analysis we did for the *Alb-R26^{Met}* model. In particular, we first identified all CpGs differentially methylated in the HBV model, then we correlated them with gene expression levels. We found a total of 115 genes both differentially methylated and differentially expressed. Notably, the total number of genes differentially methylated and differentially expressed in the HBV model is very similar to the total number of genes found in the *Alb-R26^{Met}* genetic setting (97 genes). Nevertheless, we intriguingly found a very divergent distribution, with an enrichment in genes both hypomethylated and downregulated. Intrigued by these findings, we performed correlative studies between the HBV model and the HCC cohort from TCGA (the 41 patients we report in Figure 5A in the revised version of the manuscript, new Figure 6A). Among the "NO H+E+" patient subset (18 patients), 8 patients (20%) share the same enrichment of hypomethylated and downregulated genes modelled by the HBV genetic setting. Unexpectedly, only 1/8 patients is positive for Hepatitis B virus. These results suggest that an epigenetic rewiring of gene sets through hypomethylation and downregulation occurs in a fraction of HCC patients, although it seems to be not a strict characteristic of HBV-associated patients. These findings indicate a rather exciting specificity in how genes are epigenetically reprogrammed in HCC. For example, enhanced RTK levels could ensure an enrichment in hypermethylated and upregulated genes (which act in an "oncogenic module"), whereas mechanism(s) occurring in the HBV mouse model could rather ensure an enrichment in hypomethylated and downregulated genes. We thank the reviewer for this inspiring question and we have included these finding in the revised version of the manuscript (Supplementary Figure S8). The position of the hypomethylated CGIs (promoter versus gene body) and the nature of these genes (proto-oncogenes versus tumour suppressors) still needs to be identified. In view of these results, we believe on the relevance of performing parallel and comparative studies using different HCC mouse models and on the value of such approach to decipher the heterogeneity of human HCC data. This is a topic on which we will focus our efforts in the future through collaborative studies.

4. In the Figure 2, the authors show that Decitabine treatment decreases anchorage independent growth in three analyzed cell lines and they conclude that demethylation decreases tumorigenic potential of the cells. This is too strong conclusion based only on one assay, soft agar. How about colony formation assay, proliferation, viability, senescence, cancer stem cells presence? Are they affected after Decitabine treatment?

Authors' response: We addressed this reviewer's issue with a set of experiments using two *Alb-R26^{Met}* HCC cell lines (mHCC13 and mHCC14) and two experimental conditions (Decitabine pre-treatment -03µM, 48h pre-treatment; pink in Figure 2- and Decitabine treatment during the assay - 03µM, every 2 days; green in Figure 2-). First, anchorage-dependent growth assays revealed that

Decitabine reduces both number and size of colonies. Second, tumour sphere formation assays revealed that Decitabine reduces tumour sphere number. Third, cell viability assays revealed that Decitabine treatment does not affect cell number. Collectively, these new data together with anchorage independent growth assays and xenografts in nude mice (reported before) show that DNA demethylation agents deplete tumorigenicity of HCC cells without affecting their viability (at least at the doses used for these experiments). All these data have been included in the revised version of the manuscript (Figure 2).

5. To investigate Decitabine effect in vivo, the authors use xenograft mouse model. Being that nude mice have impaired immune system (that can alter drug response), it would be more clinically relevant if for the treatment orthotopic model was used or Alb-R26Met mice with developed liver tumors. Because mouse HCC cell lines have been used, they should be able to grow in wt mice when implanted in the liver.

Authors' response: We were eager to address this issue as well and decided to test the effects of Decitabine on endogenous tumours. We got very promising results, although too preliminary to include them in the revised version of the manuscript as additional experiments are required to achieve statistically significant data. In particular, *Alb-R26^{Met}* mice with tumours were identified by non-invasive micro-computed tomography (micro-CT) imaging. Tumour volume was determined at the starting point of the experiment. Then, mice were treated either with vehicle or with Decitabine (2,5mg/kg, twice per week for a total of 5 injections). Tumour volume was determined as well after treatments. Results show that whereas tumour volume increase in mice treated with vehicle, tumour volume either remained constant or partially regressed in mice treated with Decitabine. Unfortunately, the number of available mice we had for this experiment was not sufficient to achieve statistical data. Examples of these studies and outcomes are shown in Figure 3 for reviewers.

6. After performing anchorage independent assay with the cells that have silenced Scn8a, AdamtsL5 or Actn1, the authors conclude that silencing of over-expressed genes decreases tumorigenic properties of the cells. This is strong statement based on only one assay. Additional experiments with silenced cells should be performed to solidify that conclusion, including investigating in vivo tumorigenic potential using orthotopic mouse model.

Authors' response: We agreed with the reviewer on the importance to corroborate the effect of silencing candidate genes on cell tumorigenic properties of HCC cells with further experiments. In particular, in addition to the anchorage independent growth assays, we performed: 1) anchorage dependent growth assays (also known as foci formation assay); 2) tumour sphere formation assays; 3) xenografts in nude mice. Collectively, all these studies demonstrate that downregulation of each candidate gene in *Alb-R26^{Met}* HCC cells interferes with their tumorigenic properties. All these data have been included in the revised version of the manuscript (Figure 7 and supplementary Figure S13).

We have selected 3 new candidates belonging to the list of genes both hypermethylated and overexpressed in *Alb-R26^{Met}* tumours (*Srd5a2*, *NFkB2*, and *Neur11b*). In addition to the 2 genes reported in the previous version (*Scn8a* and *Actn1*), for all the 5 genes we have generated mHCC cells stably transfected with shRNA targeting sequence (using 2 different shRNAs) and performed all functional studies mentioned above: anchorage independent assays, anchorage dependent assays, tumour sphere formation assays, and xenografts in nude mice. Collectively these studies show that each of these five genes contribute to cell tumorigenicity.

7. What causes changes in global DNA methylation in tumor cells? Is DNMT3B affected in Alb-R26Met hepatocytes?

Authors' response: Globally, the percentage of CGIs differentially methylated in *Alb-R26^{Met}* tumours compared to control livers is 3,2% (513 CGIs among the total of 16026 mouse CGIs), being 2,6% (426 CGIs) hypermethylated (Figure 1D). Following the reviewer's suggestion, we analysed the expression of the DNMTs (*DNMT1*, *DNMT3A*, and *DNMT3B*) by RNA-seq and found no significant changes ($\log_2FC < 1$) between control livers and *Alb-R26^{Met}* tumours, although with a trend of increased levels for *DNMT1*. We further analysed the expression of the DNMTs by RT-qPCR in wild-type, *Alb-R26^{Met}* healthy and *Alb-R26^{Met}* liver adjacent to tumours, as well as in *Alb-R26^{Met}* tumours. We found a significant overexpression of *DNMT1* mRNA levels in *Alb-R26^{Met}* tumours compared to wild-type and *Alb-R26^{Met}* healthy livers (whereas the difference was not significant when compared with liver adjacent to tumours). In contrast, no significant differences were observed for *DNMT3A* and *DNMT3B*. Additionally, we performed western blot analyses for both DNMT3A and DNMT3B (as protein levels could change even if mRNA levels are similar). A consistent increase in DNMT3A protein levels was observed in *Alb-R26^{Met}* tumours compared to control livers. Concerning DNMT1 and DNMT3B protein levels in *Alb-R26^{Met}* tumours compared with control livers, results indicate as well a likely increased protein levels, although we are not confident with these results and additional studies are required to support findings using different antibodies. Regarding this reviewer's issue, we also reasoned that it would be interesting to compare the expression levels of the DNMTs in the "H+E+" versus the "NO H+E+" patient subsets. Remarkably, we found a significant increased levels of both *DNMT1* and *DNMT3A* (whereas only a trend for *DNMT3B*) in the "H+E+" patient subset. All these findings are shown in Figure 3 for reviewers. Although collectively this set of data are very interesting and would be coherent with the hypermethylation found in the *Alb-R26^{Met}* HCC mouse model as well as in a subset of HCC patients, we favour not to include them in the revised version of the manuscript as more extensive studies are required to achieve solid conclusions.

8. In the line 232, the authors state that "overall survival of H+E+ versus "NO H+E+" patient subsets showed a trend of better prognosis for H+E+ patients". The same patients belong to aggressive "proliferative-progenitor subclass", that have poorer prognosis. Can authors comment on that?

Authors' response: We agree with the reviewer that this result is puzzling. Although we do not have a convincing explanation, it is possible that the presence of mutations, in addition to the H+E+ feature, aggravate the prognosis of "proliferative-progenitor subclass". Additionally, TCGA patient survival data are only available for 13/23 patients from H+E+ subset and for 15/18 patients from "NO H+E+" subset, which does not allow to perform robust correlative studies. Therefore, as only a trend was observed, we favour to remove these data from the Supplementary Figure to avoid confusion and misinterpretation. We thanks the reviewer to point this issue.

Minor points:

1. In the Figure 2B and C, if the cells are left in the culture for ten days or longer after the last pulse with Decitabine, do they recover?

Authors' response: Following the reviewer's suggestion, we left *Alb-R26^{Met}* cells for 10 days in culture after 48h with Decitabine treatment, then assessed cell tumorigenic properties by performing anchorage independent and anchorage dependent growth assays. Results show that *Alb-R26^{Met}* HCC cells recovered their tumorigenic properties as no differences were observed between control cells and cells left for 10 days in culture after 48h Decitabine treatment. These results indicate that HCC cells are capable to re-establish tumorigenicity after a transient demethylation treatment, likely by resetting methylation levels and increased levels of oncogenes. This would be coherent with finding

by Yang et al. (Cancer Cell 2014) showing that cancer cells are capable to restore acquired epigenetic modifications. Moreover, they are in agreement with our data showing that consecutive pulses of Decitabine treatments are more effective than one pulse as shown by studies reported in Figure 2. Results have been included in the Supplementary Figure S14 and in the discussion section of the revised version of the manuscript.

2. *Figure 5E, silencing efficiency should be demonstrated on the protein level, not only mRNA level as shown in the Supplemental File.*

Authors' response: We favoured to support the downregulation effects of the shRNA we used using a second shRNA targeting sequence for each gene (Supplementary Figure S13).

3. *Figure 5E, why are Scn8a, AdamtsL5 and Actn1 chosen to be investigated?*

Authors' response: We chose to functionally validate the contribution of these candidates (*Scn8a*, *Actn1*, *Srd5a2*, *NFkB2*, and *Neurl1b*) to cell tumorigenicity for two main reasons. First, among the 55 "H+E+" genes, we selected some of those overexpressed in a large proportion of HCC patients: *NFkB2* in 56%, *NEURL1B* in 46%, *SCN8A* in 41%, *ACTN1* in 22%, and *SRD5A2* in 5%. Second, whereas several hypermethylated and overexpressed genes are well-known oncogenes, such as *WT1*, *DLK1*, *TP73*, *EEF1A2*, *IGF1R*, *DKK1*, *SPOCK1*, *ITPKA*, *HOXA3*, *NOX4*, *FZD10*, *VASH2*, *GATA2*, and *SOX8*, the action of these five selected candidates as oncogenes in cancer cells (and in particular in HCC) has been less explored in previous studies. We clarify this aspect in the revised version of the manuscript.

REVIEWERS' COMMENTS:

Reviewer #1 (Remarks to the Author):

The authors have addressed the concerns raised by the reviewers.

Reviewer #2 (Remarks to the Author):

In the manuscript "Hypermethylation of gene body CpG islands predicts high dosage of functional oncogenes in liver cancer", Arechederra et al., investigated impact of DNA methylation on transcriptional switches associated with hepatocarcinogenesis. Most of my previously raised concerns have been addressed in this revised version. My only concern is that the authors did not agree on showing protein levels of Actn1, Scn8a, Srd5a2, NFkB2 and Neurl1b after silencing (raised as Minor point 2, current Figure 7 of revised manuscript). The authors opted to show, yet again, mRNA of indicated genes after silencing with different shRNA constructs. I believe that this is important and it should be included in the manuscript before publication.

Provided that the authors disclose protein levels (western blot) of Scn8a, Actn1, Srd5a2, NFkB2 and Neurl1b after silencing, I recommend accepting the manuscript for publication in Nature Communication.

Rebuttal letter to Reviewer #2.

Reviewer #2:

In the manuscript "Hypermethylation of gene body CpG islands predicts high dosage of functional oncogenes in liver cancer", Arechederra et al., investigated impact of DNA methylation on transcriptional switches associated with hepatocarcinogenesis. Most of my previously raised concerns have been addressed in this revised version. My only concern is that the authors did not agree on showing protein levels of Actn1, Scn8a, Srd5a2, NFkB2 and Neurl1b after silencing (raised as Minor point 2, current Figure 7 of revised manuscript). The authors opted to show, yet again, mRNA of indicated genes after silencing with different shRNA constructs. I believe that this is important and it should be included in the manuscript before publication.

Provided that the authors disclose protein levels (western blot) of Scn8a, Actn1, Srd5a2, NFkB2 and Neurl1b after silencing, I recommend accepting the manuscript for publication in Nature Communication.

Authors' response: The reviewer was asking to validate silencing efficiency (of shRNAs) at both protein and RNA levels, this for 5 genes that have been explored in depth. To this, in the first revision process we explained to the reviewer our choice to corroborate our validation analyses using a second shRNA targeting sequence, assessing RNA levels. A second shRNA has also successfully been used in a functional assay for the 5 genes. We believe that the use of a second shRNA is equally relevant for validation because it allows associating biological effects with a candidate gene, targeted with two independent shRNAs. Of course, we agree with reviewer 2 that measurement of protein levels by western blots is also a relevant method.

According to reviewer #2 request, we performed western blot analyses to document the protein levels in stable transfected cells carrying the shRNA sequence targeting the corresponding candidate genes compared with protein levels in control cells. We report in Figure 7A of the revised version of the manuscript panels with these western blots showing reduced protein levels of SCN8A, ACTN1, SRD5A2, NFkB2, and NEURL1B in *Alb-R26^{Met}* HCC cells carrying the shRNA targeting sequence we use (for both shRNA sequences for each gene).